# Universal Causality

**DOI:** 10.3390/e25040574

**Published:** 2023-03-27

**Authors:** Sridhar Mahadevan

**Affiliations:** Adobe Research, 345 Park Avenue, San Jose, CA 95110, USA; smahadev@adobe.com

**Keywords:** artificial intelligence, higher-order category theory, causality, machine learning, statistics

## Abstract

Universal Causality is a mathematical framework based on higher-order category theory, which generalizes previous approaches based on directed graphs and regular categories. We present a hierarchical framework called UCLA (Universal Causality Layered Architecture), where at the top-most level, causal interventions are modeled as a higher-order category over simplicial sets and objects. Simplicial sets are contravariant functors from the category of ordinal numbers Δ into sets, and whose morphisms are order-preserving injections and surjections over finite ordered sets. Non-random interventions on causal structures are modeled as face operators that map *n*-simplices into lower-level simplices. At the second layer, causal models are defined as a category, for example defining the schema of a relational causal model or a symmetric monoidal category representation of DAG models. The third layer corresponds to the data layer in causal inference, where each causal object is mapped functorially into a set of instances using the category of sets and functions between sets. The fourth homotopy layer defines ways of abstractly characterizing causal models in terms of homotopy colimits, defined in terms of the nerve of a category, a functor that converts a causal (category) model into a simplicial object. Each functor between layers is characterized by a universal arrow, which define universal elements and representations through the Yoneda Lemma, and induces a Grothendieck category of elements that enables combining formal causal models with data instances, and is related to the notion of *ground graphs* in relational causal models. Causal inference between layers is defined as a lifting problem, a commutative diagram whose objects are categories, and whose morphisms are functors that are characterized as different types of fibrations. We illustrate UCLA using a variety of representations, including causal relational models, symmetric monoidal categorical variants of DAG models, and non-graphical representations, such as integer-valued multisets and separoids, and measure-theoretic and topological models.

## 1. Introduction

Applied category theory [1] has been used to design algorithms for dimensionality reduction and data visualization [2], resolve impossibility theorems in data clustering [3] and propose schemes for knowledge representation [4]. Universal Causality (UC) is a mathematical framework based on applied higher-order category theory, which applies to graph-based [5] and non-graphical representations [6,7,8], and statistical [9] and non-statistical frameworks [10,11] (see Table 1 and Figure 1). Ordinary categories are defined as a collection of *objects* that interact pairwise through a collection of *morphisms*. Higher-order categories, such as simplicial sets [12], quasicategories [13] and *∞*-categories [14], model higher-order interactions among groups of objects, and generalize both directed graphs and ordinary categories. Our approach builds extensively on categories over *functors*. Causal interventions are defined over the functor category of simplicial objects, mapping ordinal numbers into sets or category objects. Causal inference is defined over the functor category of presheafs **Hom**${}_{C}\left(-,c\right)$, mapping an object *c* in category C into the set of morphisms into it. *Adjoint functors* define a pair of opposing functors between categories. Causal models are often characterized in terms of their underlying conditional independence structures. We model this relationship by adjoint functors between the category of conditional independence structures [15], based on algebraic representations such as *separoids* [10], and the category of causal models, defined by graphical approaches [16] or non-graphical approaches, such as integer-valued multisets [8] or measure-theoretic information fields [6,7]. We build extensively on *universal constructions*, such as colimits and limits, defined through *lifting diagrams* [17].

Over the past 150 years, causality has been studied using diverse formalisms (Table 1). While causal effects are inherently *directional*, differing from symmetric statistical models of correlation and invertible Bayesian inference, many causal discovery methods rely on querying a (symmetric) conditional independence oracle on submodels resulting from interventions on arbitrary subsets of variables (such as a *separating set* [27,28]). Abstractly, we can classify the causal representations in Table 1 using category theory in terms of their underlying *objects* and their associated *morphisms*. Causal morphisms can be algebraic, graph-based, logical, measure-theoretical, probabilistic or topological. For example, counterfactual mean embeddings [26] generalizes Rubin’s potential outcome model to reproducing kernel Hilbert spaces (RKHS), where the kernel mean map is used to embed a distribution in an RKHS, and the average treatment effect (ATE) is computed using mean maps. As Figure 1 emphasizes, UC is representation agnostic, and while it is related to category-theoretic approaches of causal DAG models that use symmetric monoidal categories [25,29,30], it differs substantially in many ways. UC introduces many novel ideas into the study of causal inference, including higher-order categorical structures based on simplicial sets and objects [12,13,14,31], adjoint functors mapping categories based on algebraic models of conditional independence [10] into actual causal models, lifting diagrams [17], and Grothendieck’s category of elements that generalizes the notion of *ground graphs* in relational causal models [32]. As we show later, any category, including symmetric monoidal categories, can be converted into simplicial objects by using the nerve functor, but its left adjoint that maps a simplicial set into a category is lossy, and preserves structure only up to n≤2-simplices. Higher-order category structures can be useful in modeing causal inference under interference [33], where the traditional stable unit treatment value assumption (SUTVA) is violated. Higher-order categories can also help model hierarchical interventions over groups of objects.

As Studeny [8] points out, Bayesian DAG models capture only a small percentage of all conditional independence structures. In particular, the space of DAG models grows exponentially in the number of variables, whereas the number of conditional independence structures grows super-exponentially proportional to the number of Boolean functions. Consequently, UC is intended to be a general framework that applies to representations that are more expressive than DAG models. In particular, UC can be used to analyze recent work on relational causal models [21,32]. The notion of a *ground graph* in relational causal models is a special case of the Grothendieck category of elements that plays a key role in the UCLA architecture. UC applies equally well to non-graphical algebraic representations that are much more expressive than DAG models, including integer-valued multisets [8], separoids [10], as well as measure-theoretic representations, such as causal information fields [6,7], that have been shown to generalize Pearl’s d-separation calculus [5].

Specifically, taking the simple example of a collider in Figure 1, in the Bayesian DAG parameterization, a well-established theoretical framework [34] specifies how to decompose the overall probability distribution into a product of local distributions. In contrast, in causal information fields [6,7], each variable is defined as a measurable space over a discrete or continuous set, and each local function is defined as a measurable function over its information field. For example, the information field $I_{C}$ for variable *C* is defined to be some measurable subset over a product σ-algebra that includes the σ-algebras $U_{A}$ and $U_{B}$ over its parents *A* and *B*, but the information field of *C* cannot depend on its own values, hence its local σ-algebra is defined as $\left\{\emptyset ,U_{C}\right\}$, where $U_{C}$ is the space of possible values of *C*. A full discussion of causal information fields is given in [7], who show it generalizes d-separation to models that include cycles and other more complex structures. Similarly, Studeny [8] proposed an algebraic framework called integer-valued multisets (imsets) for representing conditional independence structures far more expressive than DAG models. For the specific case of a DAG model G=(V,E), an imset in standard form [8] is defined as
$$u_{G}=\delta_{V}-\delta_{\emptyset }+\sum_{i\in V}\left(\delta_{{\mathrm{Pa}}_{i}}-\delta_{i\cup {\mathrm{Pa}}_{i}}\right)$$
where each $\delta_{V}$ term is the characteristic function associated with a set of variables *V*. Finally, separoids [10] is an algebraic framework for characterizing conditional independence as an abstract property, defined by a join semi-lattice equipped with a partial ordering ≤, and a ternary property ⫫ over triples of elements such that X⫫Y|Z defines the property that *X* is conditionally independent of *Y* given *Z*. It is worth pointing out that separoids are more general than the graphoid axiomatization [16] that underpins causal DAG models, since as Studeny [8] shows, graphoids are defined in terms of disjoint subsets of variables, which seriously limits their expressiveness. All these non-graphical representations can be naturally modeled within the UC framework. One of the unique aspects of UC is that causal interventions are themselves modeled as a (higher-order) category. Many approaches to causal discovery use a sequence of interventions, which naturally compose and form a category. To achieve representation independence, we model interventions as a higher-order category defined by simplicial sets and objects [12]. One strength of the simplicial objects framework for modeling causal interventions is that it enables modeling hierarchical interventions over groups of objects.

UC builds on the concept of *universal arrows* [35] to illuminate in a representation-independent manner the central abstractions employed in causal inference. Figure 2 explains this concept with an example, which also illustrates the connection between categories and graphs. For every (directed) graph *G*, there is a universal arrow from *G* to the “forgetful” functor *U* mapping the category **Cat** of all categories to **Graph**, the category of all (directed) graphs, where for any category *C*, its associated graph is defined by U(C). Intuitively, this forgetful functor “throws” away all categorical information, obliterating for example the distinction between the primitive morphisms *f* and *g* vs. their compositions g∘f, both of which are simply viewed as edges in the graph U(C). To understand this functor, consider a directed graph U(C) defined from a category *C*, forgetting the rule for composition. That is, from the category *C*, which associates to each pair of composable arrows *f* and *g*, the composed arrow g∘f, we derive the underlying graph U(C) simply by forgetting which edges correspond to elementary arrows, such as *f* or *g*, and which are composites. For example, consider a partial order as the category C, and then define U(C) as the directed graph that results from the transitive closure of the partial ordering.

The universal arrow from a graph *G* to the forgetful functor *U* is defined as a pair 〈C,u:G→U(C)〉, where *u* is a “universal” graph homomorphism. This arrow possesses the following *universal property*: for every other pair 〈D,v:G→H〉, where *D* is a category, and *v* is an arbitrary graph homomorphism, there is a functor $f^{\prime }:C\to D$, which is an arrow in the category **Cat** of all categories, such that *every* graph homomorphism ϕ:G→H uniquely factors through the universal graph homomorphism u:G→U(C) as the solution to the equation $\phi =U(f^{\prime })\circ u$, where $U\left(f^{\prime }\right):U\left(C\right)\to H$ (that is, H=U(D)). Namely, the dotted arrow defines a graph homomorphism $U(f^{\prime })$ that makes the triangle diagram “commute”, and the associated “extension” problem of finding this new graph homomorphism $U(f^{\prime })$ is solved by “lifting” the associated category arrow $f^{\prime }:C\to D$. This property of universal arrows, as we show in the paper, provide the conceptual underpinnings of universal causality in the UCLA architecture, leading to the defining property of a universal causal representation through the Yoneda Lemma [35]. Recent work on causal discovery of DAG models [27,28] can be seen as restricted ways of defining adjoint functors between causal categories of DAG models and their underlying graphs, assuming access to a conditional independence oracle that can be queried on causal sub-models resulting from interventions on arbitrary subsets of variables.

Universal causal models are defined in terms of universal constructions, such as the pullback, pushforward, (co)equalizer, and (co)limits. Figure 3 illustrates how universal causal models are functors that map from some indexing category of abstract diagrams into an actual causal model. For instance, **COVID-19 Lockdown** caused a reduction in **Traffic** and **Agricultural Fires**, which in turn caused a significant reduction in **Pollution**. In UC, we are interested in a deeper question, namely whether the pullback of **Traffic** and **Agricultural Fires** could have been some other common cause that mediated between **COVID-19 Lockdown** and its effects. If such a common cause exists, it will be viewed as a limit of an abstract causal diagram, a functor that maps from the indexing category of all diagrams to the actual causal model shown.

Figure 4 illustrates the concept of causal simplicial structures. Here, *X* denotes a causal structure represented as a category. X[0] represents the “objects” of the causal structure, defined formally as the contravariant functor X[0]:[0]→X from the simplicial category Δ to the causal category *X*. The arrows representing causal effects are defined as X[1]:[1]→X. Note that since [1]={0,1} is a category by itself, it has one (non-identity) arrow 0→1 (as well as two identity arrows). The mapping of this arrow onto *X* defines the “edges” of the causal model. Similarly, $X_{2}$ represents oriented “triangles” of three objects. Note that there is one edge from $X_{0}$ to $X_{1}$, labeled by $s_{0}$. This is a co-degeneracy operator from the simplicial layer that maps each object *A* into an identity edge **1**${}_{A}$. Similarly, there are two edges marked $d_{0}$ and $d_{1}$ from $X_{1}$ to $X_{0}$. These are co-face operators that map an edge to its source and target vertices correspondingly. Notice also that there are three edges from $X_{2}$ to $X_{1}$, marked $d_{0}$, $d_{1}$, and $d_{2}$. These are the “faces” of each 2-simplex as shown. Consider the fragment of the causal DAG model from Figure 3 shown on the right in Figure 4. The *order complex* of a DAG forms a simplicial object as shown, where the simplices are represented by the nonempty chains. In particular, each path of length *n* defines a simplex of size *n*. For example, the path from *O* (representing **Overpopulation**) to *T* (representing **Traffic**) to *P* (representing **Pollution**) defines a simplex of size 2, shown as the green shaded triangle. Note the simplices are *oriented*, which is not shown for simplicity in Figure 4. Thus, the 2-simplex formed from the chain from **O** to **T** to **P** is oriented such that O “points to” **T**, which in turn “points to” **P**. This mapping from chains over DAGs to simplicial objects is a special case of a more general construction discussed later in the paper, based on constructing the *nerve* of a category that provides a faithful functor embedding any (causal) category as a simplicial object. For example, the symmetric monoidal category representations of causal DAG models [25,29,30] can be faithfully embedded as simplicial objects by constructing their nerve.

## 2. A Layered Architecture for Universal Causality

In this paper, we propose a layered architecture that defines the framework called UCLA (Universal Causality Layered Architecture). This architecture is illustrated in Figure 5. Table 2 describes the composition of each layer. Many variants are possible, as we will discuss in the paper. As functors compose with each other, it is also possible to consider “collapsed” versions of the UCLA hierarchy.

The UCLA architecture is built on the theoretical foundation of ordinary category theory [35,36,37,38] and higher-order category theory, including quasicategories [13], and *∞*-categories [14]. As Figure 5 illustrates, at the top layer of UCLA, we model causal interventions itself as a higher-order category defined over simplicial sets and objects [12]. Causal discovery often involves a sequence of interventions, which naturally compose to form a category. Simplicial sets and simplicial objects [12] have long been a foundational framework in algebraic topology [39]. Modeling interventions using simplicial sets permits a *hierarchical* language for expressing interventions, as (co)face operators in simplicial sets and objects operate over groups of objects of arbitrary sizes. This category-theoretic approach of formalizing causal interventions gives an algebraic formalism that are related to topological notions used in causal discovery methods, such as *separating sets*[27,28] that can be defined in terms of lifting diagrams [17]. Although we will not delve into this elaboration in this paper, it is possible to define causal inference over “fuzzy” simplicial sets as well [2], which associate a real number p∈I=(0,1] with each simplicial object that denotes the uncertainty associated with a causal object or morphism. In this case, we define a fuzzy simplicial object as the functor $\Delta^{op}\times I\to C$. Fuzzy simplicial sets have been recently used in data visualization [2].

The second layer of the model represents the causal category itself, which could be a causal DAG [5], a symmetric monoidal category defining a causal DAG [25], a semi-join lattice defining a conditional independence structure, such as an integer-valued multiset [8], a relational database defining a relational causal model [21], or a causal information field [6,7], which uses a measure-theoretic notion of causality. At the third layer, we model the actual data defining a causal model by a category of instances. Finally, at the bottom-most layer, we use a homotopy category to define equivalences among causal models.

The *Grothendieck Category of Elements* (GCE) is a type of universal construction [35,37,40] that plays a central role in the UCLA architecture. It is remarkably similar to other representations widely used in database theory, and specifically in the context of causal inference, it is related to the *ground graph* used in relational causal models [21,32]. However, GCE is far more general than the ground graph construction in that it can be used to embed any object or indeed any category in **Cat**, the category of all categories.

We use *lifting diagrams* [17] to formalize causal inference at each layer of the hierarchy. A lifting problem in a category C is a morphism h:B→X in C satisfying p∘h=ν and h∘f=μ as indicated in the commutative diagram below.

                            
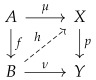


Lifting diagrams were shown to be capable of expressing SQL queries in relational databases [4]. Here, we extend this approach to model causal inference under non-random interventions, exploiting the capability of the simplicial layer to impose non-random “surgical” operations on a causal category.

Finally, to explain the bottom-most layer in UCLA of homotopy categories, it is well known that causal models are not identifiable from observations alone [5]. For example, the three distinct causal DAG models over three variables A←B→C, A→B→C and A←B←C have the same conditional independence structure, and are equivalent given a dataset of values of the variables. To model the non-distinguishability of causal models under observation, we introduce the concept of homotopic equivalence comes from topology, and is intended to reflect equivalence under some invertible mappings. A *homotopy* from a morphism f:X→Y to another morphism g:X→Y is a continuous function h:X×[0,1]→Y satisfying h(0,x)=f(x) and h(x,1)=g(x). In the category **Top** of topological spaces, homotopy defines an equivalence class on morphisms. In the application to causal inference, we can define causal homotopy [41] as finding the “quotient space” of the category of all causal models under a given set of invertible morphisms mapping one causal model into another equivalent model.

## 3. Categories, Functors, and Universal Arrows

We introduce the basic theory underlying UC in more depth now, building on relationship between categories and graphs illustrated in Figure 2. Given a graph, we can define the “free” category associated with it where we consider all possible paths between pairs of vertices (including self-loops) as the set of morphisms between them. In the reverse direction, given a category, we can define a “forgetful” functor that extracts the underlying graph from the category, forgetting the composition rule.

**Definition 1.** *A* **graph** *G (sometimes referred to as a quiver) is a labeled directed multi-graph defined by a set O of*objects*, a set A of*arrows*, along with two morphisms s:A→O and t:A→O that specify the domain and co-domain of each arrow. In this graph, we define the set of composable pairs of arrows by the set*$$A\times_{O}A=\left\{\langle g,f\rangle |\;g,f\in A,\;\;s\left(g\right)=t\left(f\right)\right\}$$*A* **category** *C is a graph G with two additional functions: id:O→A, mapping each object c∈C to an arrow ${\mathrm{id}}_{c}$ and $\circ :A\times_{O}A\to A$, mapping each pair of composable morphisms 〈f,g〉 to their composition g∘f.*

It is worth emphasizing that no assumption is made here of the finiteness of a graph, either in terms of its associated objects (vertices) or arrows (edges). Indeed, it is entirely reasonable to define categories whose graphs contain an infinite number of edges. A simple example is the group Z of integers under addition, which can be represented as a single object, denoted {•} and an infinite number of morphisms f:•→•, each of which represents an integer, where composition of morphisms is defined by addition. In this example, all morphisms are invertible. In a general category with more than one object, a *groupoid* defines a category all of whose morphisms are invertible.

As our paper focuses on the use of category theory to formalize causal inference, we interpret causal changes in terms of the concept of isomorphisms in category theory. We will elaborate this definition later in the paper.

**Definition 2.** *Two objects X and Y in a category C are deemed*  **isomorphic***, or X≅Y if and only if there is an invertible morphism f:X→Y, namely f is both* left invertible* using a morphism g:Y→X so that g∘f=* **id***${}_{X}$, and f is* right invertible *using a morphism h where f∘h=* **id***${}_{Y}$. A* **causally isomorphic change** *in a category is defined as a change of a causal object Y into $\hat{Y}$ under an intervention that changes another object X into $\hat{X}$ such that $\hat{Y}≅Y$, that is, they are isomorphic. A***causal non-isomorphic effect** *is a change that leads to a non-isomorphic change where $\hat{Y}\neg ≅Y$. An alternate definition would be to define a causally isomorphic change as a change that is an isomorphism in the category whose morphisms are causal changes.*

In the category **Sets**, two finite sets are considered isomorphic if they have the same number of elements, as it is then trivial to define an invertible pair of morphisms between them. In the category **Vect**${}_{k}$ of vector spaces over some field *k*, two objects (vector spaces) are isomorphic if there is a set of invertible linear transformations between them. As we will see below, the passage from a set to the “free” vector space generated by elements of the set is another manifestation of the universal arrow property.

Functors can be viewed as a generalization of the notion of morphisms across algebraic structures, such as groups, vector spaces, and graphs. Functors do more than functions: they not only map objects to objects, but like graph homomorphisms, they need to also map each morphism in the domain category to a corresponding morphism in the co-domain category. Functors come in two varieties, as defined below.

 **Definition 3.** *A* **covariant functor**  *F:C→D from category C to category D, and defined as the following:*

*An object FX (also written as F(X)) of the category D for each object X in category C.*

*An arrow F(f):FX→FY in category D for every arrow f:X→Y in category C.*

*The preservation of identity and composition: $F\;id_{X}=id_{FX}$ and (Ff)(Fg)=F(g∘f) for any composable arrows f:X→Y,g:Y→Z.*



**Definition 4.** *A* **contravariant functor**  *F:C→D from category C to category D is defined exactly like the covariant functor, except all the arrows are reversed.*

### 3.1. Universal Arrows

This process of going from a category to its underlying directed graph embodies a fundamental universal construction in category theory, called the *universal arrow* [35]. It lies at the heart of many useful results, principally the Yoneda Lemma that shows how object identity itself emerges from the structure of morphisms that lead into (or out of) it. The Yoneda Lemma codifies the meaning of universal causality, as it implicitly states that any change to an object must be accompanied by a change to its presheaf structure. Consequently, we can model UC in a representation-independent manner using the Yoneda Lemma.

**Definition 5.** *Given a functor S:D→C between two categories, and an object c of category C, a* **universal arrow** *from c to S is a pair 〈r,u〉, where r is an object of D and u:c→Sr is an arrow of C, such that the following universal property holds true:*

*For every pair 〈d,f〉 with d an object of D and f:c→Sd an arrow of C, there is a unique arrow $f^{\prime }:r\to d$ of D with $Sf^{\prime }\circ u=f$.*



Above we used the example of functors between graphs and their associated “free” categories and graphs to illustrate universal arrows. A central principle in the UCLA architecture is that every pair of categorical layers is synchronized by a functor, along with a universal arrow. We explore the universal arrow property more deeply in this section, showing how it provides the conceptual basis behind the Yoneda Lemma, and Grothendieck’s category of elements. In the case of causal inference, universal arrows enable mimicking the effects of causal operations from one layer of the UCLA hierarchy down to the next layer. In particular, at the simplicial object layer, we can model a causal intervention in terms of face and degeneracy operators (defined below in more detail). These in turn correspond to “graph surgery” [5] operations on causal DAGs, or in terms of “copy”, “delete” operators in “string diagram surgery” of causal models defined on symmetric monoidal categories [25]. These “surgery” operations at the next level may translate down to operations on probability distributions, measurable spaces, topological spaces, or chain complexes. This process follows a standard construction used widely in mathematics, for example group representations associate with any group *G*, a left **k**-module *M* representation that enables modeling abstract group operations by operations on the associated modular representation. These concrete representations must satisfy the universal arrow property for them to be faithful. A special case of the universal arrow property is that of universal element, which as we will see below plays an important role in the UCLA architecture in defining a suitably augmented category of elements, based on a construction introduced by Grothendieck.

**Definition 6.** *If D is a category and H:D→* 
**Set** 
*is a set-valued functor, a* 
**universal element** 
*associated with the functor H is a pair 〈r,e〉 consisting of an object r∈D and an element e∈Hr such that for every pair 〈d,x〉 with x∈Hd, there is a unique arrow f:r→d of D such that (Hf)e=x.*

**Example 1.** 
*Let E be an equivalence relation on a set S, and consider the quotient set S/E of equivalence classes, where p:S→S/E sends each element s∈S into its corresponding equivalence class. The set of equivalence classes S/E has the property that any function f:S→X that respects the equivalence relation can be written as $fs=fs^{\prime }$ whenever $s{∼}_{E}s^{\prime }$, that is, $f=f^{\prime }\circ p$, where the unique function $f^{\prime }:S/E\to X$. Thus, 〈S/E,p〉 is a universal element for the functor H.*


### 3.2. The Grothendieck Category of Elements

We turn next to define the category of elements, based on a construction by Grothendieck, and illustrate how it can serve as the basis for inference at each layer of the UCLA architecture. This definition is a special case of a general construction by Grothendieck [40].

**Definition 7.** *Given a set-valued functor δ:C→***Set** *from some category C, the induced* **Grothendieck category of elements** *associated with δ is a pair $\left(\int \delta ,\pi_{\delta }\right)$, where ∫δ∈***Cat** 
*is a category in the category of all categories* 
**Cat***, and $\pi_{\delta }:\int \delta \to C$ is a functor that “projects” the category of elements into the corresponding original category C. The objects and arrows of ∫δ are defined as follows:*

*Ob(∫δ)={(s,x)|x∈Ob(C),x∈δs}.*

**Hom**

${}_{\int \delta }\left(\left(s,x\right),\left(s^{\prime },x^{\prime }\right)\right)=\left\{f:s\to s^{\prime }|\delta \left(f\right)\left(x\right)=x^{\prime }\right\}$




**Example 2.** *To illustrate the category of elements construction, let us consider the toy climate change causal model shown in Figure 6. Let the category C be defined by this causal DAG model, where the objects Ob(C) are defined by the four vertices, and the arrows* **Hom***${}_{C}$ are defined by the four edges in the model. The set-valued functor δ:C→***Set** *maps each object (vertex) in C to a set of instances, thereby turning the causal DAG model into an associated set of tables.*

Later in the paper, we give an application of the category of elements construction to relational causal models, where in particular, it gives a rigorous semantics for ideas such as *relational skeleton* and the *ground graph* proposed in [21,32].

### 3.3. Yoneda Lemma

The Yoneda Lemma plays a crucial role in UC because it defines the concept of a representation in category theory. We first show that associated with universal arrows is the corresponding induced isomorphisms between **Hom** sets of morphisms in categories. This universal property then leads to the Yoneda Lemma.

**Theorem 1.** *Given any functor S:D→C, the universal arrow 〈r,u:c→Sr〉 implies a bijection exists between the* **Hom** *sets*$${\mathrm{Hom}}_{D}\left(r,d\right)≃{\mathrm{Hom}}_{C}\left(c,Sd\right)$$

While this is a well-known result whose proof can be found in [35], the crucial point here is its implication for causal inference. As we will see later, often in the modeling of causal inference using symmetric monoidal categories [25,29,30], a correspondence is set up between two categories, for example the symmetric monoidal category representing the structure of a causal DAG model, and the category of stochastic matrices that defines the DAG semantics. The universal arrow theorem above shows how the morphisms over the symmetric monoidal category can be synchronized with those over the stochastic matrices, enabling causal interventions to be tracked properly. A special case of this natural transformation that transforms the identity morphism **1**${}_{r}$ leads us to the Yoneda Lemma.

                          
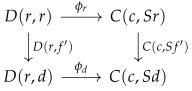


As the two paths shown here must be equal in a commutative diagram, we get the property that a bijection between the **Hom** sets holds precisely when 〈r,u:c→Sr〉 is a universal arrow from *c* to *S*. Note that for the case when the categories *C* and *D* are small, meaning their **Hom** collection of arrows forms a set, the induced functor **Hom**${}_{C}\left(c,S-\right)$ to **Set** is isomorphic to the functor **Hom**${}_{D}\left(r,-\right)$. This type of isomorphism defines a universal representation, and is at the heart of the causal reproducing property (CRP) defined below.

 **Lemma 1. Yoneda Lemma** *: If H:D→Set is a set-valued functor, and r is an object in D, there is a bijection that sends each natural transformation $\alpha :{\mathrm{Hom}}_{D}\left(r,-\right)\to K$ to $\alpha_{r}1_{r}$, the image of the identity morphism* 
**1**
*${}_{r}:r\to r$.*
$$y:\mathrm{Nat}({\mathrm{Hom}}_{D}\left(r,-\right),K)≃Kr$$

The proof of the Yoneda Lemma follows directly from the below commutative diagram, a special case of the above diagram for universal arrows.

                          
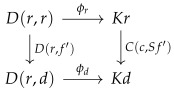


### 3.4. The Universality of Diagrams and the Causal Reproducing Property

We state two key results that underly UC, and while both these results follow directly from basic theorems in category theory, their significance for causal inference is what makes them particularly noteworthy. The first result pertains to the notion of diagrams as functors, and shows that for the functor category of presheaves, which is a universal representation of causal inference, every presheaf object can be represented as a colimit of representables through the Yoneda Lemma. This result can be seen as a generalization of the very simple result in set theory that each set is a union of one element sets. The second result is the causal reproducing property, which shows that the set of all causal effects between two objects is computable from the presheaf functor objects defined by them. Both these results are abstract, and apply to any category representation of a causal model.

Diagrams play a key role in defining UC and the UCLA architecture, as has already become clear from the discussion above. We briefly want to emphasize the central role played by universal constructions involving limits and colimits of diagrams, which are viewed as functors from an indexing category of diagrams to a category. To make this somewhat abstract definition concrete, let us look at some simpler examples of universal properties, including co-products and quotients (which in set theory correspond to disjoint unions). Coproducts refer to the universal property of abstracting a group of elements into a larger one.

Before we formally the concept of limit and colimits [35], we consider some examples. These notions generalize the more familiar notions of Cartesian products and disjoint unions in the category of **Sets**, the notion of meets and joins in the category **Preord** of preorders, as well as the least upper bounds and greatest lower bounds in lattices, and many other concrete examples from mathematics.

**Example 3.** *If we consider a small “discrete” category D whose only morphisms are identity arrows, then the colimit of a functor F:D→C is the* categorical coproduct* of F(D) for D, an object of category D, is denoted as*$${Colimit}_{D}F=\underset{D}{⨆}F\left(D\right)$$*In the special case when the category C is the category* 
**Sets**
*, then the colimit of this functor is simply the disjoint union of all the sets F(D) that are mapped from objects D∈D.*

**Example 4.** *Dual to the notion of colimit of a functor is the notion of*limit*. Once again, if we consider a small “discrete” category D whose only morphisms are identity arrows, then the limit of a functor F:D→C is the* categorical product* of F(D) for D, an object of category D, is denoted as*$${limit}_{D}F=\prod_{D}F\left(D\right)$$*In the special case when the category C is the category* 
**Sets** 
*, then the limit of this functor is simply the Cartesian product of all the sets F(D) that are mapped from objects D∈D.*

#### Pullback and Pushforward Mappings

Universal causal models in UC are defined in terms of *universal constructions*, which satisfy a universal property. We can illustrate this concept using pullback and pushforward mappings. These notions help clarify the idea of the Grothendieck category of elements, which plays a key role in the UCLA architecture.

                          
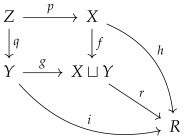


An example of a universal construction is given by the above commutative diagram, where the coproduct object X⊔Y uniquely factorizes any mapping h:X→R, such that any mapping i:Y→R, so that h=r∘f, and furthermore i=r∘g. Co-products are themselves special cases of the more general notion of co-limits. Figure 7 illustrates the fundamental property of a *pullback*, which along with *pushforward*, is one of the core ideas in category theory. The pullback square with the objects U,X,Y and *Z* implies that the composite mappings $g\circ f^{\prime }$ must equal $g^{\prime }\circ f$. In this example, the morphisms *f* and *g* represent a *pullback* pair, as they share a common co-domain *Z*. The pair of morphisms $f^{\prime },g^{\prime }$ emanating from *U* define a *cone*, because the pullback square “commutes” appropriately. Thus, the pullback of the pair of morphisms f,g with the common co-domain *Z* is the pair of morphisms $f^{\prime },g^{\prime }$ with common domain *U*. Furthermore, to satisfy the universal property, given another pair of morphisms x,y with common domain *T*, there must exist another morphism k:T→U that “factorizes” x,y appropriately, so that the composite morphisms $f^{\prime }\;k=y$ and $g^{\prime }\;k=x$. Here, *T* and *U* are referred to as *cones*, where *U* is the limit of the set of all cones “above” *Z*. If we reverse arrow directions appropriately, we get the corresponding notion of pushforward. So, in this example, the pair of morphisms $f^{\prime },g^{\prime }$ that share a common domain represent a pushforward pair. As Figure 7, for any set-valued functor δ:S→**Sets**, the Grothendieck category of elements ∫δ can be shown to be a pullback in the diagram of categories. Here, ${\mathrm{Set}}_{*}$ is the category of pointed sets, and π is a projection that sends a pointed set (X,x∈X) to its underlying set *X*.

In the category **Sets**, we know that every object (i.e., a set) *X* can be expressed as a coproduct of its elements $X≃{⊔}_{x\in X}\left\{x\right\}$, where x∈X. Note that we can view each element x∈X as a morphism x:{*}→X from the one-point set to *X*. The categorical generalization of this result is called the *density theorem* in the theory of sheaves [36]. First, we define the key concept of a *comma category*.

**Definition 8.** *Let F:D→C be a functor from category D to C. The* **comma category** *F↓C is one whose objects are pairs (D,f), where D∈D is an object of D and f∈* **Hom** *${}_{C}\left(F\left(D\right),C\right)$, where C is an object of C. Morphisms in the comma category F↓C from (D,f) to $\left(D^{\prime },f^{\prime }\right)$, where $g:D\to D^{\prime }$, such that $f^{\prime }\circ F\left(g\right)=f$. We can depict this structure through the following commutative diagram:*

                          
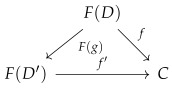


We first introduce the concept of a *dense* functor [40]:

**Definition 9.** *Let D be a small category, C be an arbitrary category, and F:D→D be a functor. The functor F is* **dense** *if for all objects C of C, the natural transformation*$$\psi_{F}^{C}:F\circ U\to \Delta_{C},\;\;{\left(\psi_{F}^{C}\right)}_{\left(D,f\right)}=f$$*is universal in the sense that it induces an isomorphism ${Colimit}_{F↓C}F\circ U≃C$. Here, U:F↓C→D is the projection functor from the comma category F↓C, defined by U(D,f)=D.*

A fundamental consequence of the category of elements is that every object in the functor category of presheaves, namely contravariant functors from a category into the category of sets, is the colimit of a diagram of representable objects, via the Yoneda Lemma. Notice this is a generalized form of the density notion from the category **Sets**, as explained above.

 **Theorem 2. Universality of Diagrams in UC** *: In the functor category of presheaves* **Set** *${}^{C^{op}}$, every object P is the colimit of a diagram of representable objects, in a canonical way [36].*

To explain the significance of this result for causal inference, note that UC represents causal diagrams as functors from an indexing category of diagrams to an actual causal model (as illustrated earlier in Figure 3). The density theorem above tells us that every presheaf object can be represented as a colimit of (simple) representable objects, namely functor objects of the form ${\mathrm{Hom}}_{C}\left(-,c\right)$.

Reproducing Kernel Hilbert Spaces (RKHS’s) transformed the study of machine learning, precisely because they are the unique subcategory in the category of all Hilbert spaces that have representers of evaluation defined by a kernel matrix K(x,y)[42]. The reproducing property in an RKHS is defined as 〈K(x,−),K(−,y)〉=K(x,y). An analogous but far more general reproducing property holds in the UC framework, based on the Yoneda Lemma. The significance of the Causal Reproducing Property is that presheaves act as “representers” of causal information, precisely analogous to how kernel matrices act as representers in an RKHS.

 **Theorem 3. Causal Reproducing Property:** 
*All causal influences between any two objects X and Y can be derived from its presheaf functor objects, namely*

$${\mathrm{Hom}}_{C}\left(X,Y\right)≃\mathrm{Nat}\left({\mathrm{Hom}}_{C}\left(-,X\right),{\mathrm{Hom}}_{C}\left(-,Y\right)\right)$$



**Proof.** The proof of this theorem is a direct consequence of the Yoneda Lemma, which states that for every presheaf functor object *F* in $\hat{C}$ of a category C, **Nat**(**Hom**${}_{C}\left(-,X\right),F)≃FX$. That is, elements of the set FX are in 1−1 bijections with natural transformations from the presheaf **Hom**${}_{C}\left(-,X\right)$ to *F*. For the special case where the functor object F=**Hom**${}_{C}\left(-,Y\right)$, we get the result immediately that **Hom**${}_{C}\left(X,Y\right)≃$**Nat**(**Hom**${}_{C}\left(-,X\right)$,**Hom**${}_{C}\left(-,Y\right))$. □

In UC, any causal influence of an object *X* upon any other object *Y* can be represented as a natural transformation (a morphism) between two functor objections in the presheaf category $\hat{C}$. The CRP is very akin to the idea of the reproducting property in kernel methods.

### 3.5. Lifting Problems

The UCLA hierarchy is defined through a series of categorical abstractions of a causal model, ranging from a combinatorial model defined by a simplicial object down to a measure-theoretic or topological realization. Between each pair of layers, we can formulate a series of lifting problems [17]. Lifting problems provide elegant ways to define basic notions in a wide variety of areas in mathematics. For example, the notion of injective and surjective functions, the notion of separation in topology, and many other basic constructs can be formulated as solutions to lifting problems. Database queries in relational databases can be defined using lifting problems [4]. Lifting problems define ways of decomposing structures into simpler pieces, and putting them back together again.

**Definition 10.** *Let C be a category. A* **lifting problem** *in C is a commutative diagram σ in C.*

                            
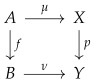


**Definition 11.** *Let C be a category. A* **solution to a lifting problem** *in C is a morphism h:B→X in C satisfying p∘h=ν and h∘f=μ as indicated in the diagram below.*

                            
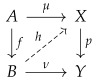


**Definition 12.** *Let C be a category. If we are given two morphisms f:A→B and p:X→Y in C, we say that f has the* **left lifting property** *with respect to p, or that p has the* **right lifting property** *with respect to f if for every pair of morphisms μ:A→X and ν:B→Y satisfying the equations p∘μ=ν∘f, the associated lifting problem indicated in the diagram below.*                            
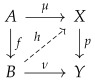

*admits a solution given by the map h:B→X satisfying p∘h=ν and h∘f=μ.*


**Example 5.** *Given the paradigmatic non-surjective morphism f:∅→{•}, any morphism p that has the right lifting property with respect to f is a* **surjective mapping** *. .*

                            
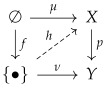


**Example 6.** *Given the paradigmatic non-injective morphism f:{•,•}→{•}, any morphism p that has the right lifting property with respect to f is an* **injective mapping** *. .*

                            
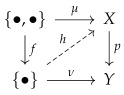


## 4. Universal Conditional Independence in Categories

Before proceeding to further detail the UCLA architecture, we discuss the special role played by conditional independence in causal inference. Causal models can be abstractly characterized by their underlying conditional independences. A number of previous axiomatizations such as *graphoids* [5,16], *integer-valued multisets* [8], and *separoids* [10] can be subsumed under the category-theoretic notion of universal conditional independence [15]. Conditional independence structures have been actively studied in AI, causal inference, machine learning, probability, and statistics for many years. Dawid [10] define separoids, a join semi-lattice, to formalize reasoning about conditional independence and irrelevance in many areas, including statistics. Pearl [16] introduced *graphoids*, a distributive lattice over disjoint subsets of variables, to model reasoning about irrelevance in probabilistic systems, and proposed representations using directed acyclic graphs (DAGs). Studeny [8] proposed a lattice-theoretic model of conditional independences using integer-valued multisets to address the intrinsic limitations of DAG-based representations.

In particular, we want to show how it is possible to define *universal conditional independence* [15], a representation of conditional independence in any category. We build specifically on the notion of *separoids* [10], an algebraic characterization of conditional independence. Recent work by Fritz and Klingler [30] has proposed a symmetric monoidal category representation of DAG type causal models, and an associated categorical probabilistic representations of *d*-separation. Our goals are to construct a more abstract representation of conditional independence based on non-graphical representations, like separoids [10] as well as integer-valued multisets [8].

Conditional independence plays a key role in causal discovery as it is often used as an oracle in causal discovery from data. Consider the problem of causal discovery as inferring a directed acyclic graph (DAG) G=(V,E) from data, where the conditional independence ⊥−10mu⊥ property is defined using the graph property of *d*-separation [16]. A given DAG *G* can be characterized in two ways: one parameterization specifies the DAG *G* in terms of the vertices *V* and edges *E*, which corresponds to specifying the objects and morphisms of a category defining the DAG. The second way to parameterize a DAG is by its induced collection of conditional independence properties, as defined by *d*-separation. For example, the serial DAG over three variables, A→B→C, can be defined using its two edges A→B and B→C, but also by its conditional independences, namely A⊥−10mu⊥C|B using the theory of *d*-separation. We are thus given two possibly redundant parameterizations of the same algebraic structure. However, multiple DAG models can define the same conditional independences. For example, the serial model A→B→C, as well as the “diverging” model A←B→C and the “reverse” serial model A←B←C all capture the same conditional independence property (A⊥−10mu⊥C|B). This non-uniqueness property arises because Bayes rule can be used to map any one of these three DAGs into the form represented by one of the other DAGs.

### 4.1. The Category of Separoids

A separoid (S,≤,⫫)[10] is defined as a semi-lattice S, where the join ∨ operator over the semi-lattice S defines a preorder ≤, and the ternary relation ⫫ is defined over triples of the form (x⫫y|z) (which are interpreted to mean *x* is conditionally independent of *y* given *z*). We show briefly how to define a category for universal conditional independence, where each object is a separoid, and the morphisms are homomorphisms from one separoid to another. It is possible to define “lattice” objects in any category by interpreting an arrow f:x→y as defining the partial ordering [36].

**Definition 13.** *A* **separoid** *[10] defines a category over a preordered set (S,≤), namely ≤ is reflexive and transitive, equipped with a*ternary*relation ⫫ on triples (x,y,z), where x,y,z∈S satisfy the following properties:*
**S1:** 
*(S,≤) is a join semi-lattice.***P1:** 
*x⫫y|x*
**P2:** 
*x⫫y|z⇒y⫫x|z*
**P3:** 
*x⫫y|zandw≤y⇒x⫫w|z*
**P4:** 
*x⫫y|zandw≤y⇒x⫫y|(z∨w)*
**P5:** 
*x⫫y|zandx⫫w|(y∨z)⇒x⫫(y∨w)|z*

*A* 
**strong separoid** 
*also defines a categoroid. A strong separoid is defined over a lattice S has in addition to a join ∨, a meet ∧ operation, and satisfies an additional axiom:*
**P6** 
*: If z≤y and w≤y, then x⫫y|zandx⫫y|w⇒x⫫y|z∧w*


To define a category of separoids, we have to define the notion of a homomorphism between separoids [10]:

**Definition 14.** *Let 〈S,≤,⫫〉 and $\langle S^{\prime },\leq^{\prime },{⫫}^{\prime }\rangle$ be two separoids. A map $f:S\to S^{\prime }$ is a* 
**separoid homomorphism** 
*if:*
(1)
*It is a join-lattice homomorphism, namely $f\left(x\vee y\right)=f\left(x\right)\vee^{\prime }f\left(y\right)$, which implies that $x\leq y\to f\left(x\right)\leq^{\prime }f\left(y\right)$.*
(2)
*$x⫫y\;|z\to f\left(x\right){⫫}^{\prime }f\left(y\right)\;|f\left(z\right)$.*
(3)
*In case both S and S’ are strong separoids, we can define the notion of a strong separoid homomorphism to additionally include the condition: $f\left(x\wedge y\right)\to f\left(x\right)\wedge^{\prime }f\left(y\right)$.*



With this definition, we can now define the category of separoids and a representation-independent characterization of universal conditional independence as follows:

**Theorem 4.** 
*The category of separoids is defined as one where each object in the category is defined as a separoid 〈S,≤,⫫〉, and the arrows are defined as (strong) separoid homomorphisms. The category of separoids provides an axiomatization of universal conditional independence, namely that it enables a universal representation through the use of universal arrows and Yoneda Lemma.*


**Proof.** First, we note that the category of separoids indeed forms a category as it straightforwardly satisfies all the basic properties. The (strong) separoid homomorphisms compose, so that g∘f as a composition of two (strong) separoid homomorphisms produces another (strong) separoid homomorphism. The universal property derives from the use of the Yoneda Lemma to define a category of presheaves that map from the category of separoids to the category **Sets**. □

### 4.2. Adjoint Functors in Causal Discovery

First, we need to review the basic concept of adjoint functors, which will be helpful in modeling several aspects of causal inference in this paper.

**Definition 15.** *A pair of* 
**adjoint functors** 
*is defined as F:C→D and G:D→C, where F is considered the right adjoint, and G is considered the left adjoint,*                            
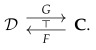

*must satisfy the property that for each pair of objects C of C and D of D, there is a natural transformation between the two sets of morphisms*

$$\phi_{C,D}:{\mathrm{Hom}}_{C}\left(C,G\left(D\right)\right)≃{\mathrm{Hom}}_{D}\left(F\left(C\right),D\right)$$



An important property of adjoint functors is connected to the concepts of limits and colimits reviewed above.

**Theorem 5.** 
*If F and G are a pair of adjoint functors*
                            
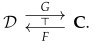

*then the functor G preserves colimits and the functor F preserves limits.*


Notice the similarity of this definition to the one earlier where the universal arrow property induced a bijection of **Hom** sets that then led to universal elements, Grothendieck category of elements, and the Yoneda Lemma.

We now introduce the perspective of adjoint functors for causal discovery (see Figure 8). Many causal discovery algorithms [27] that use a conditional independence oracle to query conditional independence properties from a dataset can be viewed in this perspective as using adjoint functors between the category of separoids and the category of the causal model itself. We can design functors that map from the category of all separoids into the category of causal models (in particular, for example, the category of graphs, or the category of integer-valued multisets [8]). Shown in the figure is one particular separoid object with a single conditional independence property stating that *A* and *B* are dependent conditional on knowing the value of *C*), which can realized in two ways: one using a collider DAG A→C←B, and the other as a integer-valued multiset. These pair of functors are an example of the general case of adjoint functors between “forgetful” and “free” functors [40]. To make this more precise, let us define the “forgetful” functor *R* between a causal model on the right to its underlying set of conditional independences on the left, so that R(M) is the separoid object that represents the conditional independence in a causal model *M*. Note that *R* is a “forgetful” functor, in that it “throws away” structural information, including for example, whether the causal model is a causal DAG or an integer-valued multiset. On the other hand, the “free” functor L(M), its left adjoint, maps a given separoid object to any of its associated “free” objects, namely causal models that represent it, irrespective of their formalism. Within the category of causal models, morphisms enable translation between different representations.

## 5. Layers 1 and 2: Category of Causal Interventions over Simplicial Objects

We now discuss Layers 1 and 2 in UCLA architecture, describing the top simplicial objects layer, and how it interacts with the causal category structure (layer 2). Simplicial sets are higher-dimensional generalizations of directed graphs, partially ordered sets, as well as regular categories themselves. Importantly, simplicial sets and simplicial objects form a foundation for higher-order category theory [13,14]. By using simplicial sets and objects at the top layer, UCLA enables a powerful machinery to define a higher-order category for representing a rich class of causal interventions over a very expressive set of causal models, including relational causal models [32], and perform abstract “diagram surgery”, for example “graph surgery” [5] or “string diagram surgery” [25].

Simplicial objects have long been a foundation for algebraic topology [12,39], and more recently in higher-order category theory [13,14,43]. The category Δ has non-empty ordinals [n]={0,1,…,n] as objects, and order-preserving maps [m]→[n] as arrows. An important property in Δ is that any many-to-many mapping is decomposable as a composition of an injective and a surjective mapping, each of which is decomposable into a sequence of elementary injections $\delta_{i}:\left[n\right]\to \left[n+1\right]$, called *coface* mappings, which omits i∈[n], and a sequence of elementary surjections $\sigma_{i}:\left[n\right]\to \left[n-1\right]$, called *co-degeneracy* mappings, which repeats i∈[n]. The fundamental simplex Δ([n]) is the presheaf of all morphisms into [n], that is, the representable functor Δ(−,[n]). The Yoneda Lemma [35] assures us that an *n*-simplex $x\in X_{n}$ can be identified with the corresponding map Δ[n]→X. Every morphism f:[n]→[m] in Δ is functorially mapped to the map Δ[m]→Δ[n] in S.

Any morphism in the category Δ can be defined as a sequence of *co-degeneracy* and *co-face* operators, where the co-face operator $\delta_{i}:\left[n-1\right]\to \left[n\right],0\leq i\leq n$ is defined as:$$\delta_{i}\left(j\right)=\left\{\begin{array}{cc} j, & for\;0\leq j\leq i-1\\ j+1 & for\;i\leq j\leq n-1 \end{array}\right.$$

Analogously, the co-degeneracy operator $\sigma_{j}:\left[n+1\right]\to \left[n\right]$ is defined as
$$\sigma_{j}\left(k\right)=\left\{\begin{array}{cc} j, & for\;0\leq k\leq j\\ k-1 & for\;j<k\leq n+1 \end{array}\right.$$

Note that under the contravariant mappings, co-face mappings turn into face mappings, and co-degeneracy mappings turn into degeneracy mappings. That is, for any simplicial object (or set) $X_{n}$, we have $X\left(\delta_{i}\right)d_{i}:X_{n}\to X_{n-1}$, and likewise, $X\left(\sigma_{j}\right)s_{j}:X_{n-1}\to X_{n}$.

The compositions of these arrows define certain well-known properties [12,40]:$$\begin{array}{ccc} \delta_{j}\circ \delta_{i} & = & \delta_{i}\circ \delta_{j-1},\;\;i<j\\ \sigma_{j}\circ \sigma_{i} & = & \sigma_{i}\circ \sigma_{j+1},\;\;i\leq j\\ \sigma_{j}\circ \delta_{i}\left(j\right) & = & \left\{\begin{array}{cc} \sigma_{i}\circ \sigma_{j+1}, & for\;i<j\\ 1_{\left[n\right]} & for\;i=j,j+1\\ \sigma_{i-1}\circ \sigma_{j},for\;i>j+1 \end{array}\right. \end{array}$$

**Example 7.** 
*The “vertices” of a simplicial object $C_{n}$ are the objects in C, and the “edges” of C are its arrows f:X→Y, where X and Y are objects in C. Given any such arrow, the degeneracy operators $d_{0}f=Y$ and $d_{1}f=X$ recover the source and target of each arrow. Also, given an object X of category C, we can regard the face operator $s_{0}X$ as its identity morphism $1_{X}:X\to X$.*


**Example 8.** 
*Given a category C, we can identify an n-simplex σ of a simplicial set $C_{n}$ with the sequence:*

$$\sigma =C_{o}\overset{f_{1}}{\to }C_{1}\overset{f_{2}}{\to }\ldots \overset{f_{n}}{\to }C_{n}$$

*the face operator $d_{0}$ applied to σ yields the sequence*

$$d_{0}\sigma =C_{1}\overset{f_{2}}{\to }C_{2}\overset{f_{3}}{\to }\ldots \overset{f_{n}}{\to }C_{n}$$

*where the object $C_{0}$ is “deleted” along with the morphism $f_{0}$ leaving it. The “edge intervention” model in [44] effectively can be viewed as deleting the vertex from which the edge originates.*


**Example 9.** 
*Given a category C, and an n-simplex σ of the simplicial set $C_{n}$, the face operator $d_{n}$ applied to σ yields the sequence*

$$d_{n}\sigma =C_{0}\overset{f_{1}}{\to }C_{1}\overset{f_{2}}{\to }\ldots \overset{f_{n-1}}{\to }C_{n-1}$$

*where the object $C_{n}$ is “deleted” along with the morphism $f_{n}$ entering it. Note this face operator can be viewed as analogous to interventions on leaf nodes in a causal DAG model.*


**Example 10.** 
*Given a category C, and an n-simplex σ of the simplicial set $C_{n}$ the face operator $d_{i},0<i<n$ applied to σ yields the sequence*

$$d_{i}\sigma =C_{0}\overset{f_{1}}{\to }C_{1}\overset{f_{2}}{\to }\ldots C_{i-1}\overset{f_{i+1}\circ f_{i}}{\to }C_{i+1}\ldots \overset{f_{n}}{\to }C_{n}$$

*where the object $C_{i}$ is “deleted” and the morphisms $f_{i}$ is composed with morphism $f_{i+1}$. Note that this process can be abstractly viewed as intervening on object $C_{i}$ by choosing a specific value for it (which essentially “freezes” the morphism $f_{i}$ entering object $C_{i}$ to a constant value).*


**Example 11.** 
*Given a category C, and an n-simplex σ of the simplicial set $C_{n}$, the degeneracy operator $s_{i},0\leq i\leq n$ applied to σ yields the sequence*

$$s_{i}\sigma =C_{0}\overset{f_{1}}{\to }C_{1}\overset{f_{2}}{\to }\ldots C_{i}\overset{1_{C_{i}}}{\to }C_{i}\overset{f_{i+1}}{\to }C_{i+1}\ldots \overset{f_{n}}{\to }C_{n}$$

*where the object $C_{i}$ is “repeated” by inserting its identity morphism $1_{C_{i}}$.*


**Definition 16.** 
*Given a category C, and an n-simplex σ of the simplicial set $C_{n}$, σ is a*
**degenerate**
*simplex if some $f_{i}$ in σ is an identity morphism, in which case $C_{i}$ and $C_{i+1}$ are equal.*


### 5.1. Simplicial Subsets and Horns

We now describe more complex ways of extracting parts of causal structures using simplicial subsets and horns. These structures will play a key role in defining suitable lifting problems.

**Definition 17.** *The* 
**standard simplex** 
*$\Delta^{n}$ is the simplicial set defined by the construction*
$$\left(\left[m\right]\in \Delta \right)\mapsto {\mathrm{Hom}}_{\Delta }\left(\left[m\right],\left[n\right]\right)$$
*By convention, $\Delta^{-1}\emptyset$. The standard 0-simplex $\Delta^{0}$ maps each $\left[n\right]\in \Delta^{op}$ to the single element set {•}.*


**Definition 18.** *Let $S_{•}$ denote a simplicial set. If for every integer n≥0, we are given a subset $T_{n}\subseteq S_{n}$, such that the face and degeneracy maps*$$d_{i}:S_{n}\to S_{n-1}\;\;\;\;s_{i}:S_{n}\to S_{n+1}$$*applied to $T_{n}$ result in*$$d_{i}:T_{n}\to T_{n-1}\;\;\;\;s_{i}:T_{n}\to T_{n+1}$$*then the collection ${\left\{T_{n}\right\}}_{n\geq 0}$ defines a* **simplicial subset**   *$T_{•}\subseteq S_{•}$*


**Definition 19.** *The* 
**boundary** 
*is a simplicial set $\left(\partial \Delta^{n}\right):\Delta^{op}\to$* 
**Set** 
*defined as*
$$\left(\partial \Delta^{n}\right)\left(\left[m\right]\right)=\left\{\alpha \in {\mathrm{Hom}}_{\Delta }\left(\left[m\right],\left[n\right]\right):\alpha \;is\;not\;surjective\right\}$$

Note that the boundary $\partial \Delta^{n}$ is a simplicial subset of the standard *n*-simplex $\Delta^{n}$.

**Definition 20.** *The* **Horn** *$\Lambda_{i}^{n}:\Delta^{op}\to$* **Set** 
*is defined as*
$$\left(\Lambda_{i}^{n}\right)\left(\left[m\right]\right)=\left\{\alpha \in {\mathrm{Hom}}_{\Delta }\left(\left[m\right],\left[n\right]\right):\left[n\right]\neg \subseteq \alpha \left(\left[m\right]\right)\cup \left\{i\right\}\right\}$$

Intuitively, the Horn $\Lambda_{i}^{n}$ can be viewed as the simplicial subset that results from removing the interior of the *n*-simplex $\Delta^{n}$ together with the face opposite its *i*th vertex.

### 5.2. Example: Causal Intervention and Horn Filling of Simplicial Objects

Let us illustrate this abstract discussion above by instantiating it in the context of causal inference. Figure 9 instantiates the abstract discussion above in terms of an example from causal inference. We are given a simple 3 variable DAG, on which we desire to explore the causal effect of variable *A* on *C*. Using Pearl’s backdoor criterion, we can intervene on variable *A* by freezing its value do(A=1), for example, which will eliminate the dependence of *A* on *B*. Consider now the lifting problem where we want to know if there is a completion of this simplicial subset $\Lambda_{2}^{2}$, which is a “outer horn”.

We can view the causal intervention problem in the more abstract setting of a class of lifting problem, shown with the following diagrams. Consider the problem of composing 1-dimensional simplices to form a 2-dimensional simplicial object. Each simplicial subset of an *n*-simplex induces a a *horn* $\Lambda_{k}^{n}$, where 0≤k≤n. Intuitively, a horn is a subset of a simplicial object that results from removing the interior of the *n*-simplex and the face opposite the *i*th vertex. Consider the three horns defined below. The dashed arrow ⇢ indicates edges of the 2-simplex $\Delta^{2}$ not contained in the horns.

            
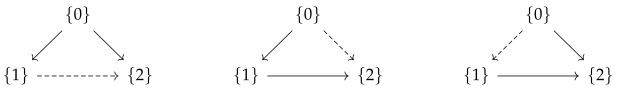


The inner horn $\Lambda_{1}^{2}$ is the middle diagram above, and admits an easy solution to the “horn filling” problem of composing the simplicial subsets. The two outer horns on either end pose a more difficult challenge. For example, filling the outer horn $\Lambda_{0}^{2}$ when the morphism between {0} and {1} is *f* and that between {0} and {2} is the identity 1 is tantamount to finding the left inverse of *f* up to homotopy. Dually, in this case, filling the outer horn $\Lambda_{2}^{2}$ is tantamount to finding the right inverse of *f* up to homotopy. A considerable elaboration of the theoretical machinery in category theory is required to describe the various solutions proposed, which led to different ways of defining higher-order category theory [13,14,43].

### 5.3. Higher-Order Categories

We now formally introduce higher-order categories, building on the framework proposed in a number of formalisms [13,14,43]. We briefly summarize various approaches to the horn filling problem in higher-order category theory.

**Definition 21.** *Let f:X→S be a morphism of simplicial sets. We say f is a* **Kan fibration** *if, for each n>0, and each 0≤i≤n, every lifting problem.*            
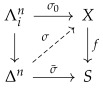

*admits a solution. More precisely, for every map of simplicial sets $\sigma_{0}:\Lambda_{i}^{n}\to X$ and every n-simplex $\bar{\sigma }:\Delta^{n}\to S$ extending $f\circ \sigma_{0}$, we can extend $\sigma_{0}$ to an n-simplex $\sigma :\Delta^{n}\to X$ satisfying $f\circ \sigma =\bar{\sigma }$.*


**Example 12.** *Given a simplicial set X, then a projection map $X\to \Delta^{0}$ that is a Kan fibration is called a* **Kan complex**
.


**Example 13.** 
*Any isomorphism between simplicial sets is a Kan fibration.*


**Example 14.** 
*The collection of Kan fibrations is closed under retracts.*


**Definition 22** ([14]). *An ∞-category is a simplicial object $S_{•}$ which satisfies the following condition:*

*For 0<i<n, every map of simplicial sets $\sigma_{0}:\Lambda_{i}^{n}\to S_{•}$ can be extended to a map $\sigma :\Delta^{n}\to S_{i}$.*



This definition emerges out of a common generalization of two other conditions on a simplicial set $S_{i}$:**Property K**: For n>0 and 0≤i≤n, every map of simplicial sets $\sigma_{0}:\Lambda_{i}^{n}\to S_{•}$ can be extended to a map $\sigma :\Delta^{n}\to S_{i}$.**Property C**: for 0<1<n, every map of simplicial sets $\sigma_{0}:\Lambda_{i}^{n}\to S_{i}$ can be extended uniquely to a map $\sigma :\Delta^{n}\to S_{i}$.

Simplicial objects that satisfy property K were defined above to be Kan complexes. Simplicial objects that satisfy property C above can be identified with the nerve of a category, which yields a full and faithful embedding of a category in the category of sets. Definition 22 generalizes both of these definitions, and was called a *quasicategory* in [13] and *weak Kan complexes* in [43] when C is a category. We will use the nerve of a category below in defining homotopy colimits as a way of characterizing a causal model.

### 5.4. Example: Simplicial Objects over Integer-Valued Multisets

To help ground out this somewhat abstract discussion above on simplicial objects and sets, let us consider its application to two other examples. Our first example comes from a non-graphical representations of conditional independence, namely integer-valued multisets [8], defined as an integer-valued multiset function $u:Z^{P(Z)}\to Z$ from the power set of integers, P(Z) to integers Z. An imset is defined over partialy ordered set (poset), defined as a distributive lattice of disjoint (or non-disjoint) subsets of variables. The bottom element is denoted *∅*, and top element represents the complete set of variables *N*. A full discussion of the probabilistic representations induced by imsets is given [8]. We will only focus on the aspects of imsets that relate to its conditional independence structure, and its topological structure as defined by the poset. A *combinatorial* imset is defined as:$$u=\sum_{A\subset N}c_{A}\delta_{A}$$
where $c_{A}$ is an integer, $\delta_{A}$ is the characteristic function for subset *A*, and *A* potentially ranges over all subsets of *N*. An *elementary* imset is defined over (a,b⊥−10mu⊥A), where a,b are singletons, and A⊂N∖{a,b}. A *structural* imset is defined as one where the coefficients can be rational numbers. For a general DAG model G=(V,E), an imset in standard form [8] is defined as
$$u_{G}=\delta_{V}-\delta_{\emptyset }+\sum_{i\in V}\left(\delta_{{\mathrm{Pa}}_{i}}-\delta_{i\cup {\mathrm{Pa}}_{i}}\right)$$

The space of all possible imset representations over *n* variables defines a lattice [8], where the top of the lattice corresponds to the “discrete" causal model with no non-trivial morphisms, and the bottom of the lattice corresponds to the complete model with morphisms between every pair of objects. Each candidate imset defines a *causal horn*, a simplicial subobject of the complete simplex, and the process of causal structure discovery can be viewed in terms of the abstract horn filling problem defined above for higher-order categories.

### 5.5. Example: Simplicial Objects over String Diagrams

We now illustrate the above formalism of simplicial objects by illustrating how it applies to the special case where causal models are defined over symmetric monoidal categories [25,29,30]. For a detailed overview of symmetric monoidal categories, we recommend the book-length treatment by Fong and Spivak [1]. Symmetric monoidal categories (SMCs) are useful in modeling processes where objects can be combined together to give rise to new objects, or where objects disappear. For example, Coecke et al. [45] propose a mathematical framework for resources based on SMCs. We focus on the work of Jacobs et al. [25]. It is important to point out that monoidal categories can be defined as a special type of Grothendieck fibration [40]. We discuss one specific case of the general Grothendieck construction in the next section construction, and refer the reader to [40] for how the structure of monoidal categories itself emerges from this construction.

Our goal in this section is to illustrate how we can define simplicial objects over the SMC category CDU category **Syn**${}_{G}$ constructed by Jacobs et al. [25] to mimic the process of working with an actual Bayesian network DAG *G* For the purposes of our illustration, it is not important to discuss the intricacies involved in this model, for which we refer the reader to the original paper. Our goal is to show that by encapsulating their SMC category in the UCLA framework, we can extend their approach as described below. In particular, we can solve an associated lifting problem that is defined by the functor mapping the simplicial category Δ to their SMC category. They use the category of stochastic matrices to capture the process of working with the joint distribution as shown in the figure. Instead, we show that one can use some other category, such as the category of **Sets**, or **Top** (the category of topological spaces), or indeed, the category **Meas** of measurable spaces.

Recall that Bayesian networks [16] define a joint probability distribution
$$P\left(X_{1},\ldots ,X_{n}\right)=\prod_{i=1}^{n}P\left(X_{i}|Pa\left(X_{i}\right)\right],$$
where $Pa\left(X_{i}\right)\subset \left\{X_{1},\ldots ,X_{n}\right\}\setminus X_{i}$ represents a subset of variables (not including the variable itself). Jacobs et al. [25] show Bayesian network models can be constructed using symmetric monoidal categories, where the tensor product operation is used to combine multiple variables into a “tensored” variable that then probabilistically maps into an output variable. In particular, the monoidal category **Stoch** has as objects finite sets, and morphisms f:A→B are |B|×|A| dimensional stochastic matrices. Composition of stochastic matrices corresponds to matrix multiplication. The monoidal product ⊗ in **Stoch** is the cartesian product of objects, and the Kronecker product of matrices f⊗g. Jacobs et al. [25] define three additional operations, the copy map, the discarding map, and the uniform state.

**Definition 23.** 
*A CDU category (for copy, discard, and uniform) is a SMC category (*
**C**
*, *⊗*, I), where each object A has a copy map $C_{A}:A\to A⊗A$, and discarding map $D_{A}:A\to I$, and a uniform state map $U_{A}:I\to A$, satisfying a set of equations detailed in Jacobs et al. [25]. CDU functors are symmetric monoidal functors between CDU categories, preserving the CDU maps.*


The key theorem we are interested in is the following from the original paper [25]:

**Theorem 6.** *There is an isomorphism (1-1 correspondence) between Bayesian networks based on a DAG G and CDU functors F:* **Syn** *${}_{G}\to$***Stoch**
.


The significance of this theorem for the UCLA architecture is that it shows how the SMC category of CDU objects can be defined as Layer 2 of the UCLA hierarchy, whereas the category **Stoch** can be viewed as instantiating the Layer 3 of the UCLA hierarchy. Notice that this theorem in effect defines a universal arrow between the CDU category and the category of stochastic matrices, which is a central unifying principle in UC.

### 5.6. Nerve of a Category

An important concept that will play a key role in Layer 4 of the UCLA hierarchy is that of the *nerve* of a category [40]. The nerve of a category C enables embedding C into the category of simplicial objects, which is a fully faithful embedding.

**Definition 24.** 
*Let F:C→D be a functor from category C to category D. If for all arrows f the mapping f→Ff*

*injective, then the functor F is defined to be* **faithful**
.
*surjective, then the functor F is defined to be* **full**
.
*bijective, then the functor F is defined to be* **fully faithful**
.



**Definition 25.** *The* **nerve** 
*of a category C is the set of composable morphisms of length n, for n≥1. Let $N_{n}\left(C\right)$ denote the set of sequences of composable morphisms of length n.*
$$\left\{C_{o}\overset{f_{1}}{\to }C_{1}\overset{f_{2}}{\to }\ldots \overset{f_{n}}{\to }C_{n}\;|\;C_{i}\;is\;an\;object\;in\;C,f_{i}\;is\;a\;morphism\;in\;C\right\}$$

The set of *n*-tuples of composable arrows in C, denoted by $N_{n}\left(C\right)$, can be viewed as a functor from the simplicial object [n] to C. Note that any nondecreasing map α:[m]→[n] determines a map of sets $N_{m}\left(C\right)\to N_{n}\left(C\right)$. The nerve of a category C is the simplicial set $N_{•}:\Delta \to N_{n}\left(C\right)$, which maps the ordinal number object [n] to the set $N_{n}\left(C\right)$.

The importance of the nerve of a category comes from a key result [40], showing it defines a full and faithful embedding of a category:

**Theorem 7.** *The* **nerve functor** *$N_{•}:$* 
**Cat** 
*→* 
**Set** 
*is fully faithful. More specifically, there is a bijection θ defined as:*
$$\theta :\mathrm{Cat}\left(C,C^{\prime }\right)\to {\mathrm{Set}}_{\Delta }(N_{•}\left(C\right),N_{•}\left(C^{\prime }\right)$$

Using this concept of a nerve of a category, we can now state a theorem that shows it is possible to easily embed the CDU symmetric monoidal category defined above that represents Bayesian Networks and their associated “string diagram surgery” operations for causal inference as a simplicial set.

**Theorem 8.** *Define the* **nerve** *of the CDU symmetric monoidal category* (**C**, ⊗, *I*), *where each object A has a copy map $C_{A}:A\to A⊗A$, and discarding map $D_{A}:A\to I$, and a uniform state map $U_{A}:I\to A$ as the set of composable morphisms of length n, for n≥1. Let $N_{n}\left(C\right)$ denote the set of sequences of composable morphisms of length n.*
$$\left\{C_{o}\overset{f_{1}}{\to }C_{1}\overset{f_{2}}{\to }\ldots \overset{f_{n}}{\to }C_{n}\;|\;C_{i}\;is\;an\;object\;in\;C,f_{i}\;is\;a\;morphism\;in\;C\right\}$$*The associated* 
**nerve functor** 
*$N_{•}:$*
**Cat** *→* **Set** *from the CDU category is fully faithful. More specifically, there is a bijection θ defined as:*
$$\theta :\mathrm{Cat}\left(C,C^{\prime }\right)\to {\mathrm{Set}}_{\Delta }(N_{•}\left(C\right),N_{•}\left(C^{\prime }\right)$$

This theorem is just a special case of the above theorem attesting to the full and faithful embedding of any category using its nerve, which then makes it a simplicial set. We can then use the theoretical machinery at the top layer of the UCLA architecture to manipulate causal interventions in this category using face and degeneracy operators as defined above.

Note that the functor *G* from a simplicial object *X* to a category C can be lossy. For example, we can define the objects of C to be the elements of $X_{0}$, and the morphisms of C as the elements $f\in X_{1}$, where f:a→b, and $d_{0}f=a$, and $d_{1}f=b$, and $s_{0}a,a\in X$ as defining the identity morphisms $1_{a}$. Composition in this case can be defined as the free algebra defined over elements of $X_{1}$, subject to the constraints given by elements of $X_{2}$. For example, if $x\in X_{2}$, we can impose the requirement that $d_{1}x=d_{0}x\circ d_{2}x$. Such a definition of the left adjoint would be quite lossy because it only preserves the structure of the simplicial object *X* up to the 2-simplices. The right adjoint from a category to its associated simplicial object, in contrast, constructs a full and faithful embedding of a category into a simplicial set. In particular, the nerve of a category is such a right adjoint.

## 6. Layers 2 and 3 of UCLA: The Category of Elements in Causal Inference

Next, we turn to describe the second (from top) and third layers of the UCLA architecture, which pertain to the category of causal models (for example, a graph or a symmetric monoidal category), and the database of instances that support causal inferences. Drawing on the close correspondences between between categories and relational database schemes (see [4] for details), we can view causal queries over data as analogous to database queries, which can then be formulated by corresponding lifting problems. That is, each object in the model, e.g., a variable indicating a patient, maps into actual patients, and a variable indicating outcomes from COVID-19 exposure, maps into actual outcomes for that individual. The causal arrow from the patient variable into the exposure variable then maps into actual arrows for each patient. Causal queries of exposure to COVID-19 then become similar to database queries. In the next section, we will generalize this perspective, showing that we can map into a topological category and answer more abstract questions relating to the geometry of a dataset, or map into a category of measurable spaces to answer probabilistic queries. The structure of the lifting problem remains the same, what changes are the specifics of the underlying categories.

### 6.1. Grothendieck Category of Elements in Relational Causal Models

The Grothendieck category of elements is related to the notion of ground graphs used in relational causal models [32]. Using the example in their papers, we are given three generic objects, **Employee**, **Product**, and **Business-Unit**, and several morphisms, including **Develops**, **Funds**, **Salary**, **Competence**, **Revenue** and **Budget**. We can view a relational schema as shown as a category, following the approach shown in [46,47]. Note each object, such as **Employee**, maps using a functor into the category **Sets** into actual employees, such as **Paul** or **Sally**. Each morphism in the category, for example **Develops** must accordingly also be mapped by this functor into a set-valued function. So, as illustrated, we have that **Sally** is involved in developing a **Laptop**, and **Paul** is involved in developing a **Case**, both of which of course are instances of **Product**. The GCE for this relational causal model is strongly related to the so-called *relational skeleton* and *ground graph* explored in relational causal models [21,32].

A full discussion of these connections is beyond the scope of this paper, but there are some interesting differences to be noted. In their approach, relations such as **Develops** are depicted as undirected, whereas in our case, we model these as directional properties (which seems natural in this example). Ahsan et al. [32] develop a notion of *relational d-separation* in their work, and it would be interesting to construct a categorified version of that notion, an interesting problem for future work. We turn instead to discuss how GCE plays a key role in lifting problems associated with causal inference in UCLA. These provide a rigorous semantics to their use in relational causal models as well, which might be a fruitful avenue to explore in subsequent work.

### 6.2. Lifting Problems in Causal Inference

Many properties of Grothendieck’s construction can be exploited (some of these are discussed in the context of relational database queries in [4]), but for our application to causal inference, we are primarily interested in the associated class of lifting problems that define queries in a causal model.

**Definition 26.** *If S is a collection of morphisms in category C, a morphism f:A→B has the* **left lifting property with respect to S** *if it has the left lifting property with respect to every morphism in S. Analogously, we say a morphism p:X→Y has the* **right lifting property with respect to S** *if it has the right lifting property with respect to every morphism in S.*

We now turn to sketch some examples of the application of lifting problems for causal inference. Many problems in causal inference on graphs involve some particular graph property. To formulate it as a lifting problem, we will use the following generic template, following the initial application of lifting problems to database queries proposed by Spivak [4].

            
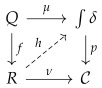


Here, *Q* is a generic query that we want answered, which could range from a database query, as in the original setting studied by Spivak [4], but more interestingly, it could be a particular graph property relating to causal inference (as illustrated by the following two examples), but as we will show later, it could also be related to the combinatorial category of simplicial objects used to model causal intervention, and finally, it could also be related to questions relating to the evaluation of causal models using a measure-theoretic or probability space. By suitably modifying the base category, the lifting problem formulation can be used to encode a diverse variety of problems in causal inference. *R* represents a fragment of the complete causal model C, and δ is the category of elements defined above. Finally, *h* gives all solutions to the lifting problem. Some examples will help clarify this concept.

**Example 15.** *Consider the category of directed graphs defined by the category G, where Ob(G) = {V, E}, and the morphisms of G are given as* **Hom***${}_{G}$ = {s, t}, where s:E→V and t:E→V define the source and terminal nodes of each vertex. Then, the category of all directed graphs is precisely defined by the category of all functors δ:G→***Set***. Any particular graph is defined by the functor X:G→***Set***, where the function X(s):X(E)→X(V) assigns to every edge its source vertex. For causal inference, we may want to check some property of a graph, such as the property that every vertex in X is the source of some edge. The following lifting problem ensures that every vertex has a source edge in the graph. The category of elements ∫δ shown below refers to a construction introduced by Grothendieck, which will be defined in more detail later.*

         
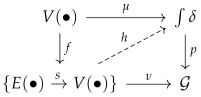


**Example 16.** 
*As another example of the application of lifting problems to causal inference, let us consider the problem of determining whether two causal DAGs, $G_{1}$ and $G_{2}$ are Markov equivalent [48]. A key requirement here is that the immoralities of $G_{1}$ and $G_{2}$ must be the same, that is, if $G_{1}$ has a collider A→B←C, where there is no edge between A and C, then $G_{2}$ must also have the same collider, and none others. We can formulate the problem of finding colliders as the following lifting problem. Note that the three vertices A, B and C are bound to an actual graph instance through the category of elements ∫δ (as was illustrated above), using the top right morphism μ. The bottom left morphism f binds these three vertices to some collider. The bottom right morphism ν requires this collider to exist in the causal graph G with the same bindings as found by μ. The dashed morphisms h finds all solutions to this lifting problem, that is, all colliders involving the vertices A, B and C.*


        
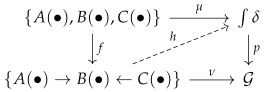


If the category of elements is defined by a functor mapping a database schema into a table of instances, then the associated lifting problem corresponds to familiar problems like SQL queries in relational databases [4]. In our application, we can use the same machinery to formulate causal inference queries by choosing the categories appropriately. To complete the discussion, we now make the connection between universal arrows and the core notion of universal representations via the Yoneda Lemma.

### 6.3. Modeling Causal Interventions as Kan Extension

It is well known in category theory that ultimately every concept, from products and co-products, limits and co-limits, and ultimately even the Yoneda Lemma (see below), can be derived as special cases of the Kan extension [35]. Kan extensions intuitively are a way to approximate a functor F so that its domain can be extended from a category C to another category D. Because it may be impossible to make commutativity work in general, Kan extensions rely on natural transformations to make the extension be the best possible approximation to F along K. We want to briefly show Kan extensions can be combined with the category of elements defined above to construct causal “migration functors” that map from one causal model into another. These migration functors were originally defined in the context of database migration [4], and here we are adapting that approach to causal inference. By suitably modifying the category of elements from a set-valued functor δ:C→**Set**, to some other category, such as the category of topological spaces, namely δ:C→ **Top**, we can extend the causal migration functors into solving more abstract causal inference questions. We explore the use of such constructions in the next section on Layer 4 of the UCLA hierarchy. Here, for simplicity, we restrict our focus to Kan extensions for migration functors over the category of elements defined over instances of a causal model.

**Definition 27.** *A* **left Kan extension** *of a functor F:C→E along another functor K:C→D, is a functor ${Lan}_{K}F:D\to E$ with a natural transformation $\eta :F\to {Lan}_{F}\circ K$ such that for any other such pair (G:D→E,γ:F→GK), γ factors uniquely through η. In other words, there is a unique natural transformation $\alpha :{Lan}_{F}\Rightarrow G$.*

            
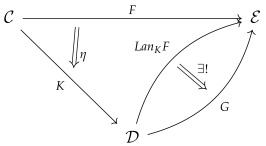


A **right Kan extension** can be defined similarly. To understand the significance of Kan extensions for causal inference, we note that under a causal intervention, when a causal category *S* gets modified to *T*, evaluating the modified causal model over a database of instances can be viewed as an example of Kan extension.

Let δ:S→**Set** denote the original causal model defined by the category *S* with respect to some dataset. Let ϵ:T→**Set** denote the effect of a causal intervention abstractly defined as some change in the category *S* to *T*, such as deletion of an edge, as illustrated in Figure 10. Intuitively, we can consider three cases: the *pullback* $\Delta_{F}$ along *F*, which maps the effect of a causal intervention back to the original model, the *left pushforward* $\Sigma_{F}$ and the *right pushforward* $\prod_{F}$, which can be seen as adjoints to the pullback $\Delta_{F}$.

Following [4], we can define three *causal migration functors* that evaluate the impact of a causal intervention with respect to a dataset of instances.

The functor $\Delta_{F}:\epsilon \to \delta$ sends the functor ϵ:T→**Set** to the composed functor δ∘F:S→ **Set**.The functor $\Sigma_{F}:\delta \to \epsilon$ is the left Kan extension along *F*, and can be seen as the left adjoint to $\Delta_{F}$.The functor $\prod_{F}:\delta \to \epsilon$ is the right Kan extension along *F*, and can be seen as the right adjoint to $\Delta_{F}$.

To understand how to implement these functors, we use the following proposition that is stated in [4] in the context of database queries, which we are restating in the setting of causal inference.

**Theorem 9.** *Let F:S→T be a functor. Let δ:S→***Set** *and ϵ:T→***Set** *be two set-valued functors, which can be viewed as two instances of a causal model defined by the category S and T. If we view T as the causal category that results from a causal intervention on S (e.g., deletion of an edge), then there is a commutative diagram linking the category of elements between S and T.*

            
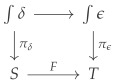


**Proof.** To check that the above diagram is a pullback, that is, $\int \delta ≃S\times_{T}\int \delta$, or in words, the fiber product, we can check the existence of the pullback component wise by comparing the set of objects and the set of morphisms in ∫δ with the respective sets in $S\times_{T}\int \epsilon$. □

For simplicity, we defined the migration functors above with respect to an actual dataset of instances. More generally, we can compose the set-valued functor δ:S→Set with a functor T:**Set**→ **Top** to the category of topological spaces to derive a Kan extension formulation of the definition of a causal intervention. We discuss this issue in the next section on causal homotopy.

## 7. Layer 4 of UCLA: Causal Homotopy

Finally, we turn to discuss the role of the causal homotopy layer. To understand the reason for considering homotopy in causal inference, note that causal models can only be determined up to some equivalence class from data, and while many causal discovery algorithms assume arbitrary interventions can be carried out, e.g., on separating sets [27], to discover the unique structure, such interventions are generally impossible to do in practical applications. The concept of *essential graph* [48] and *chain graph* [49] are attempts to formulate the notion of a “quotient space” of graphs, but similar issues arise more generally for non-graph based models as well. Thus, it is useful to understand how to formulate the notion of equivalent classes of causal models in an arbitrary category. For example, given the conditional independence structure A⫫B|C, there are at least three different symmetric monoidal categorical representations that all satisfy this conditional independence [25,29,30], and we need to define the quotient space over all such equivalent categories.

In our previous work on causal homotopy [41], we exploited the connection between causal DAG graphical models and finite topological spaces [50,51]. In particular, for a DAG model G=(V,E), it is possible to define a finite space topology T=(V,O), whose open sets O are subsets of the vertices *V* such that each vertex *x* is associated with an open set $U_{x}$ defined as the intersection of all open sets that contain *x*. This structure is referred to an *Alexandroff* topology [52]. An intuitive way to construct an Alexandroff topology is to define the open set for each variable *x* by the set of its ancestors $A_{x}$, or by the set of its descendants $D_{x}$. This approach transcribes a DAG graph into a finite topological space, upon which the mathematical tools of algebraic topology can be applied to construct homotopies among equivalent causal models. Our approach below generalizes this construction to simplicial objects, as well as general categories.

### 7.1. The Category of Fractions: Localizing Invertible Morphisms in a Category

A principal challenge in causal discovery is that models can be inferred from data only up to an equivalence class. We can view the morphisms between equivalent causal models as “invertible” arrows, which defines a construction called an “essential” graph [48]. The problem of defining a category with a given subclass of invertible morphisms, called the category of fractions [53], is another concrete illustration of the close relationships between categories and graphs. It is also useful in the context of causal inference, as for example, in defining the Markov equivalence class of directed acyclic graphs (DAGs) as a category that is localized by considering all invertible arrows as isomorphisms. Borceux [54] has a detailed discussion of the “calculus of fractions”, namely how to define a category where a subclass of morphisms are to be treated as isomorphisms. The formal definition is as follows:

**Definition 28.** *Consider a category C and a class Σ of arrows of C. The* **category of fractions*** $C(\Sigma^{-1})$ is said to exist when a category $C(\Sigma^{-1})$ and a functor $\phi :C\to C(\Sigma^{-1})$ can be found with the following properties:*

*∀f,ϕ(f) is an isomorphism.*

*If D is a category, and F:C→D is a functor such that for all morphisms f∈Σ, F(f) is an isomorphism, then there exists a unique functor $G:C(\Sigma^{-1})\to D$ such that G∘ϕ=F.*



A detailed construction of the category of fractions is given in [54], which uses the underlying directed graph skeleton associated with the category. The characterization of the Markov equivalent class of acyclic directed graphs is an example of the abstract concept of category of fractions [48]. Briefly, this condition states that two acyclic directed graphs are Markov equivalent if and only if they have the same skeleton and the same immoralities. In our previous work [41], we explored constructing homotopically invariant causal models over finite Alexandroff topological spaces, which can be seen as a special case of the UCLA framework since finite topological (Alexandroff) spaces define a category [52].

### 7.2. Homotopy of Simplicial Objects

We will discuss homotopy in categories more generally now. This notion of homotopy generalizes the notion of homotopy in topology, which defines why an object like a coffee cup is topologically homotopic to a doughnut (they have the same number of “holes”).

**Definition 29.** 
*Let C and $C^{\prime }$ be a pair of objects in a category C. We say C is*
**a retract**
*of $C^{\prime }$ if there exists maps $i:C\to C^{\prime }$ and $r:C^{\prime }\to C$ such that $r\circ i={id}_{C}$.*


**Definition 30.** *Let C be a category. We say a morphism f:C→D is a* **retract of another morphism** *$f^{\prime }:C\to D$ if it is a retract of $f^{\prime }$ when viewed as an object of the functor category* 
**Hom** 
*([1],C). A collection of morphisms T of C is* **closed under retracts** *if for every pair of morphisms $f,f^{\prime }$ of C, if f is a retract of $f^{\prime }$, and $f^{\prime }$ is in T, then f is also in T.*

**Definition 31.** *Let X and Y be simplicial sets, and suppose we are given a pair of morphisms $f_{0},f_{1}:X\to Y$. A* **homotopy** *from $f_{0}$ to $f_{1}$ is a morphism $h:\Delta^{1}\times X\to Y$ satisfying $f_{0}{=h|}_{0\times X}$ and $f_{1}=h_{1\times X}$.*

### 7.3. Singular Homology

Our goal is to define an abstract notion of a causal model in terms of its underlying classifying space as a category, and show how it can be useful in defining causal homotopy. We will also clarify how it relates to determining equivalences among causal models, namely homotopical invariance, and also how it sheds light on causal identification. First, we need to define more concretely the topological *n*-simplex that provides a concrete way to attach a topology to a simplicial object. Our definitions below build on those given in [14]. For each integer *n*, define the topological space $|\Delta_{n}|$ realized by the object $\Delta_{n}$ as
$$|\Delta_{n}|=\left\{t_{0},t_{1},\ldots ,t_{n}\in R^{n+1}:t_{0}+t_{1}+\ldots +t_{n}=1\right\}$$

This is the familiar *n*-dimensional simplex over *n* variables. For any causal model, its classifying space $|N_{•}\left(C\right)|$ defines a topological space. We can now define the *singularn*-simplex as a continuous mapping $\sigma :|\Delta_{N}\left|\to \right|N_{•}\left(C\right)|$. Every singular *n*-simplex σ induces a collection of n−1-dimensional simplices called *faces*, denoted as
$$d_{i}\sigma \left(t_{0},\ldots ,t_{n-1}\right)=\left(t_{0},t_{1},\ldots ,t_{i-1},0,t_{i},\ldots ,t_{n-1}\right)$$

Note that as discussed above, a causal intervention on a variable in a DAG can be modeled as applying one of these degeneracy operators $d_{i}$. The above definition shows that every such intervention has an effect on the topology associated with the causal model. Define the set of all morphisms ${Sing}_{n}\left(X\right)={\mathrm{Hom}}_{\mathrm{Top}}(\Delta_{n},|N_{•}\left(C\right)\left|\right)$ as the set of singular *n*-simplices of $|N_{•}\left(C\right)|$.

**Definition 32.** *For any topological space defined by a causal model $|N_{•}\left(C\right)|$, the* **singular homology groups** *$H_{*}\left(\right|N_{•}\left(C\right)\left|;Z\right)$ are defined as the homology groups of a chain complex*$$\ldots \overset{\partial }{\to }Z({Sing}_{2}\left(\right|N_{•}\left(C\right)|))\overset{\partial }{\to }Z({Sing}_{1}\left(\right|N_{•}\left(C\right)|))\overset{\partial }{\to }Z({Sing}_{0}\left(\right|N_{•}\left(C\right)|))$$*where $Z({Sing}_{n}\left(\right|N_{•}\left(C\right)|))$ denotes the free Abelian group generated by the set ${Sing}_{n}\left(\right|N_{•}\left(C\right)\left|\right)$ and the differential ∂ is defined on the generators by the formula*$$\partial \left(\sigma \right)=\sum_{i=0}^{n}{\left(-1\right)}^{i}d_{i}\sigma$$

Intuitively, a chain complex builds a sequence of vector spaces that can be used to construct an algebraic invariant of a causal model from its classifying space by choosing the left **k** module Z to be a vector space. Each differential *∂* then becomes a linear transformation whose representation is constructed by modeling its effect on the basis elements in each $Z({Sing}_{n}\left(X\right))$.

**Example 17.** 
*Let us illustrate the singular homology groups defined by an integer-valued multiset [8] used to model conditional independence. Imsets over a DAG of three variables N={a,b,c} can be viewed as a finite discrete topological space. For this topological space X, the singular homology groups $H_{*}\left(X;Z\right)$ are defined as the homology groups of a chain complex*

$$Z\left({Sing}_{3}\left(X\right)\right)\overset{\partial }{\to }Z\left({Sing}_{2}\left(X\right)\right)\overset{\partial }{\to }Z\left({Sing}_{1}\left(X\right)\right)\overset{\partial }{\to }Z\left({Sing}_{0}\left(X\right)\right)$$

*where $Z({Sing}_{i}\left(X\right))$ denotes the free Abelian group generated by the set ${Sing}_{i}\left(X\right)$ and the differential ∂ is defined on the generators by the formula*

$$\partial \left(\sigma \right)=\sum_{i=0}^{4}{\left(-1\right)}^{i}d_{i}\sigma$$


*The set ${Sing}_{n}\left(X\right)$ is the set of all morphisms ${\mathrm{Hom}}_{Top}\left(\right|\Delta_{n}\left|,X\right)$. For an imset over the three variables N={a,b,c}, we can define the singular n-simplex σ as:*

$$\sigma :|\Delta^{4}\left|\to X\;\;where\;\;\right|\Delta^{n}|=\left\{t_{0},t_{1},t_{2},t_{3}\in {\left[0,1\right]}^{4}:t_{0}+t_{1}+t_{2}+t_{3}=1\right\}$$


*The n-simplex σ has a collection of faces denoted as $d_{0}\sigma ,d_{1}\sigma ,d_{2}\sigma$ and $d_{3}\sigma$. If we pick the k-left module Z as the vector space over real numbers R, then the above chain complex represents a sequence of vector spaces that can be used to construct an algebraic invariant of a topological space defined by the integer-valued multiset. Each differential ∂ then becomes a linear transformation whose representation is constructed by modeling its effect on the basis elements in each $Z({Sing}_{n}\left(X\right))$. An alternate approach to constructing a chain homology for an integer-valued multiset is to use Möbius inversion to define the chain complex in terms of the nerve of a category (see our recent work on categoroids [15] for details).*


### 7.4. Classifying Spaces and Homotopy Colimits

Building on the intuition proposed above, we now introduce a formal way to define causal effects in our framework, which relies on the construction of a topological space associated with the nerve of a category. As we saw above, the nerve of a category is a full and faithful embedding of a category as a simplicial object.

**Definition 33.** *The* **classifying space** *of a causal model defined as a category C is the topological space associated with the nerve of the category $|N_{•}C|$.*

To understand the classifying space $|N_{•}C|$ of a causal model defined as a category C, let us go over some simple examples to gain some insight.

**Example 18.** 
*For any set X, which can be defined as a discrete category $C_{X}$ with no non-trivial morphisms, the classifying space $|N_{•}C_{X}|$ is just the discrete topology over X (where the open sets are all possible subsets of X).*


**Example 19.** 
*If we take a causal model defined as a partially ordered set [n], with its usual order-preserving morphisms, then the nerve of [n] is isomorphic to the representable functor δ(−,[n]), as shown by the Yoneda Lemma, and in that case, the classifying space is just the topological space $\Delta_{n}$ defined above.*


**Example 20.** *In our earlier work on causal homotopy [41], we associated with any finite causal DAG G, a finite Alexandroff topological space, where the open sets of the topology corresponding to the* down *sets or* upsets *of descendants or ancestors, respectively. Since any causal DAG model G induces a partial ordering, we can then define the classifying space of a causal DAG in terms of the topological space associated with the nerve of the DAG, namely $|N_{•}G|$.*

**Example 21.** *Witsenhausen [6] defined a measure-theoretic notion of causality called the*intrinsic model*. An intrinsic model $M={\left(\alpha ,U_{\alpha },I_{\alpha }\right)}_{\alpha \in A}$, where the parameters of the intrinsic causal model over n variables A are defined in terms of a collection of measurable functions over each variable’s information field $I_{\alpha }$ (a subfield of the product σ-algebra over all variables upon which it depends), where $U_{\alpha }$ is the space over which α takes its values. Heymann et al. [7] showed recently that Witsenhausen’s intrinsic model generalizes Pearl’s d-separation condition, and can be used to define a rich set of causal models that includes cycles and feedback, as well as more refined notions of conditional d-separation. The definition of causality in an intrinsic model is based on structuring the information fields of every variable in such a way that it is possible to sequentially order them for any particular instance of the underlying sample space. It is possible to define a topology on the underlying variables (which Witsenhausen referred to as*agents*), by defining*subystem*of variables B⊆A such that every variable α∈B has an information field that only depends on the information fields of members in its subset B, that is ∀α∈B, the condition states that $I_{\alpha }\subseteq F_{B}$, where $F_{B}$ is the induced product information field over the subset of variables B. Witsenhausen proves that the collection of subsystems forms a finite topology on A. We can then define the classifying space of an intrinsic causal model to be the topological space associated with the nerve of an intrinsic model M, namely $|N_{•}M|$.*

We now want to bring in the set-valued functor mapping each causal category C to the actual experiment used, e.g., in a clinical trial [9], to evaluate average treatment effect or quantify the effect of a **do** calculus intervention [5] We can then compute the topological space prior to intervention, and subsequent to intervention, and compare the two topological spaces in terms of their algebraic invariants (e.g., the chain complex, as described below).

**Definition 34.** *The* **homotopy colimit** *of a causal model defined as nerve of the category of elements associated with the set-valued functor δ:C→* 
**Set** *mapping the causal category C to a dataset, namely $N_{•}\left(\int \delta \right)$.*

In general, we may want to evaluate the homotopy colimit of a causal model not only with respect to the data used in a causal experiment, but also with respect to some underlying topological space or some measurable space. We can extend the above definition straightforwardly to these cases using an appropriate functor T: **Set**→**Top**, or alternatively M: **Set**→ **Meas**. These augmented constructions can then be defined with respect to a more general notion called the *homotopy colimit* [40] of a causal model.

**Definition 35.** *The* **topological homotopy colimit**  *${hocolim}_{T\circ \delta }$ of a causal model associated with a category C, along with its associated category of elements associated with a set-valued functor δ:C→* 
**Set**
*, and a topological functor T:* 
**Set** 
*→* 
**Top** 
*is isomorphic to topological space associated with the nerve of the category of elements, that is ${hocolim}_{T\circ \delta }≃\left|N_{•}\left(\int \delta \right)\right|$.*

**Example 22.** 
*The classifying space $|N_{•}C_{CDU}|$ associated with CDU symmetric monoidal category encoding of a causal Bayesian DAG is defined using the monoidal category (*
**C**
*, *⊗*, I), where each object A has a copy map $C_{A}:A\to A⊗A$, and discarding map $D_{A}:A\to I$, and a uniform state map $U_{A}:I\to A$, is defined as the topological realization of its nerve. As before, the nerve $N_{n}\left(C\right)$ of the CDU category is defined as the set of sequences of composable morphisms of length n.*

$$\left\{C_{o}\overset{f_{1}}{\to }C_{1}\overset{f_{2}}{\to }\ldots \overset{f_{n}}{\to }C_{n}\;|\;C_{i}\;is\;an\;object\;in\;C,f_{i}\;is\;a\;morphism\;in\;C\right\}$$

*Note that the CDU category was associated with a CDU functor F:***Syn***${}_{G}\to$* 
**Stoch** *to the category of stochastic matrices. We can now define the homotopy colimit ${hocolim}_{F}$ of the CDU causal model associated with the CDU category C, along with its associated category of elements associated with a set-valued functor δ:C→*
**Set**
*, and a topological functor F:* 
**Set** 
*→* 
**Stoch** *is isomorphic to topological space associated with the nerve of the category of elements over the composed functor, that is ${hocolim}_{F\circ \delta }$.*

### 7.5. Defining Causal Effect

Finally, we turn to defining causal effect using the notion of classifying space and homotopy colimits, as defined above. Space does not permit a complete discussion of this topic, but the basic idea is that once a causal model is defined as a topological space, there are a large number of ways of comparing two topological spaces from analyzing their chain complexes, or using a topological data analysis method such as UMAP [2].

**Definition 36.** *Let the classifying space under “treatment” be defined as the topological space $|N_{•}C_{1}|$ associated with the nerve of category $C_{1}$ under some intervention, which may result in a topological deformation of the model (e.g., deletion of an edge). Similarly, the classifying space under “no treatment” be defined as the $|N_{•}C_{0}|$ under a no-treatment setting, with no intervention. A* **causally non-isomorphic effect** *exists between categories $C_{1}$ and $C_{0}$, or $C_{1}\neg ≅C_{0}$ if and only if there is no invertible morphism $f:|N_{•}C_{1}|\to N_{•}\left(C_{0}\right|$ between the “treatment” and “no-treatment” topological spaces, namely f must be both*left invertible*and*right invertible.

There is an equivalent notion of causal effect using the homotopy colimit definition proposed above, which defines the nerve functor using the category of elements. This version is particularly useful in the context of evaluating a causal model over a dataset.

**Definition 37.** *Let the homotopy colimit ${hocolim}_{1}=\left|N_{•}\left(\int \delta_{1}\right)\right|$ be the topological space associated with a causal category $C_{1}$ under the “treatment’ condition be defined with respect to an associated category of elements defined by a set-valued functor $\delta_{1}:C\to$* **Set** *over a dataset of “treated” variables, and corresponding “no-treatment” ${hocolim}_{0}=\left|N_{•}\left(\int \delta_{0}\right)\right|$ be the topological space of a causal model associated with a category $C_{0}$ be defined over an associated category of elements defined by a set-valued functor $\delta_{0}:C\to$* **Set** *over a dataset of “placebo” variables. A* **causally non-isomorphic effect** *exists between categories $C_{1}$ and $C_{0}$, or $C_{1}\neg ≅C_{0}$ if and only if there is no invertible morphism $f:|N_{•}\left(\int \delta_{1}\right)\left|\to \right|N_{•}\left(\delta_{0}\right)|$ between the “treatment” and “no-treatment” homotopy colimit topological spaces, namely f must be both*left invertible*and*right invertible.

We can define an equivalent “do-calculus” like version of the causal effect definitions above for the case when a causal model defined as a graph structure is manipulated by an intervention that deletes an edge, or does some more sophisticated type of “category” surgery.

## 8. Contributions of Our Paper

We summarize the principal contributions of our paper. Our principal contribution is the development of the notion of “universal causality”, a representation-independent definition whose goal is to elucidate the “universal’ properties of causal inference. Our work is inspired by other work, for example separoids [10] elucidates the concept of conditional independence in a representation-independent manner, which applies to conditional independence in probability theory, statistics, and geometry. Another example is the concept of *Grothendieck topology* [36], which defines topology abstractly in the context of any category. Implicit in these constructions is the abstraction of a specific construct—conditional independence or topology–in a manner that lets it be studied across a wide range of representations. Similarly, UC is intended to be an abstract characterization of causality.

**Universal Arrow:** We used universal arrows as a unifying principle in UC, which allows synchronizing causal changes at different levels of the UCLA hierarchy. Universal arrows set up a correspondence between a “forgetful” functor and its left adjoint“free” functor. In the application to causal inference, universal arrows, for example, define forgetful and free functors between the category of conditional independence structures, such as separoids, from the category of actual causal models (e.g., as symmetric monoidal categories of causal DAG models [25,29,30]).**Causal reproducing property:** The universal arrow property leads to the powerful Yoneda Lemma, which provides the foundational result embodied in the causal reproducing property. The CRP implies that all causal influences between two objects *X* and *Y* in a category C are representable in the functor category of presheaves, namely
$${\mathrm{Hom}}_{C}\left(X,Y\right)≃\mathrm{Nat}\left({\mathrm{Hom}}_{C}\left(-,X\right),{\mathrm{Hom}}_{C}\left(-,Y\right)\right)$$**Causal interventions as a higher-order category:** Most causal discovery algorithms require a sequence of interventions, which naturally compose to form a category. We introduced the framework of higher-order category theory using simplicial sets and objects to define a category over causal interventions. Simplicial objects provides an elegant and general way of extracting parts of a compositional structure, and its associated lifting problems define when a partial fragment of a causal model can be “put back” together into a complete model.**Nerve of a causal model:** We used the nerve construction to set up a functor between a casual category and its associated simplicial object, which is a fully faithful embedding of any category as a simplicial object. Its left adjoint functor, which maps a simplicial set into a category, is a lossy representation that only preserves structure up to n≤2 simplices. Simplicial sets suggest a way to define higher-order causal models, a topic for future work.**Relational causal models:** The Grothendieck category of elements is closely related to the notion of *ground graphs* in relational causal models [32], which gives a rich source of applications of causal inference. Any relational database defines a category [4], and our paper shows how to formulate causal inference in the rich space of relational enterprise datasets.**Lifting Problem:** Associated with each pair of layers of the UCLA hierarchy is a lifting problem over a suitable category of elements, from simplicial category of elements, to a category of elements over a dataset, to a category of elements over a topological space. In general, the Grothendieck category of elements is a way to embed each object in a category into the category of all categories **Cat**. This construction has many elegant properties, which deserves further exploration in a subsequent paper.**Homotopy colimits and Classifying Spaces:** We defined causal effect in terms of the classifying space associated with the nerve of a causal category, and with the homotopy colimit of the nerve of the category of elements. These structures have been extensively explored in the study of homotopy in category theory [40], and there are many advanced techniques that can be brought to bear on this problem, such as *model categories* [55].

## 9. Future Work

There are many directions for future work, and we summarize a few of them below.

**Higher-order causality:** Our use of simplicial sets and objects suggests a way of defining higher-order causality, as simplicial sets generalize directed graphs, categories, and partial orders. Simplicial sets permit modeling the interaction between groups of objects, which naturally applies to cases of causal inference with *interference*, where the stable unit treatment value assumption (SUTVA) [9] is violated. Zigler and Papadogeorgou [33] explores an application to causal interference, where the treatment units (e.g., power plants) and response units (e.g., people living close to power plants) have a complex set of interactions, where a particular treatment may affect many individuals. These types of problems can be studied using higher-order degeneracy operators over oriented *n*-simplexes.**Causal Discovery from Conditional Independence Oracles:** The problem of causal discovery can then be rigorously formulated as a lifting problem as well, where the conditional independence oracle is defined as a solution to a lifting problem. More specifically, it is possible to define a Grothendieck category of elements for a functor *F*: **Graph**→ **Separoids** mapping the category of directed graphs into the category of separoids, which define its equivalent set of conditional independence statements. The Grothendieck fibration in this case maps the category of elements, combining conditional independence properties and graph objects, into the category **Graph**. Algorithms proposed in the literature, such as [27], can be seen as queries in a lifting problem, analogous to the lifting problems defined for the UCLA hierarchy. This approach can be extended to causal discovery over higher-order categories.**Grothendieck Topology:** Analogous to the representation-independent definition of conditional independence using separoids, our longstanding goal has been to define causality purely in terms of a categorical structure. The Grothendieck topology J for any category, which leads to the concept of a *site*[36], is defined such that for any object *c* in C, a *sieveS* is a family of morphisms, all with co-domain *c* such that
f∈S→f∘g∈S
for any *g* where the composition is defined. A Grothendieck topology J on category C defines a sieve J(c) for each object *c* such that the following properties hold: (i) the maximal sieve $t_{c}=\left\{f|cod\left(f\right)=c\right\}$ is in J(c). There is an additional stability condition and a transitive closure condition. An interesting problem for future work is to define causal inference over sheaves of a site, using the concept of Grothendieck topologies. Any causal intervention that, for instance, deletes an edge, would change the Grothendieck topology embodied in the structure of sieves.**Gröbner Causal Models:** Another direction for future work is to construct Gröbner representations of causal categories. Sam and Snowden [56] define a general **Gröbner** representation for combinatorial categories, which apply to causal models as well. Specifically, denote **Rep**${}_{k}\left(C\right)$ as the category of representations of a causal model C, where **k** is a non-zero ring, and Mod${}_{k}$ is the category of left-**k** modules. Thus, we can define a **representation** of a causal category C as a functor $C\to {Mod}_{k}$. Let *x* be an object of C. Define a representation $P_{x}$ of C as a left **k**-module, where $P_{x}\left(y\right)=k\left[{\mathrm{Hom}}_{C}\left(x,y\right)\right]$, that is, $P_{x}\left(y\right)$ is the free left **k**-module with basis **Hom**${}_{C}\left(x,y\right)$. For any particular morphism f:x→y, let $e_{f}$ denote the corresponding element of $P_{x}\left(y\right)$. Broadly speaking, this approach generalizes the work on modeling graphical models as algebraic varieties [20,57,58], and ideals on partially ordered sets (posets) [59]. The intuitive idea is that a representation of a category can be defined as an abstract Gröbner basis over an ideal defined on a module whose basis is defined using the free algebra generated by the set of all morphisms out of an object. This approach provides an alternative way of parameterizing causal models defined as combinatorial categories.

## 10. Summary

In this paper, we proposed a framework called Universal Causality (UC) for causal inference using the tools of category theory. Specifically, we described a layered hierarchical architecture called UCLA (Universal Causality Layered Architecture), where causal inference is modeled at multiple levels of categorical abstraction. At the top-most level, causal inference is modeled using a higher-order category of simplicial sets and objects, defined as contravariant functors from the category of ordinal numbers Δ, whose objects are the ordered natural numbers [n]={0,…,n}, and whose morphisms are order-preserving injections and surjections. Causal “surgery” is then modeled as the action of a contravariant functor from the category Δ into a causal model. At the second layer, causal models are defined by a category consisting of a collection of objects, such as the entities in a relational database, and morphisms between objects can be viewed as attributes relating entities. The third categorical abstract layer corresponds to the data layer in causal inference, where each causal object is mapped into a set of instances, modeled using the category of sets and morphisms are functions between sets. The fourth layer comprises of additional structure imposed on the instance layer above, such as a topological space, a measurable space or a probability space, or more generally, a locale. Between every pair of layers in UCLA are functors that map objects and morphisms from the domain category to the co-domain category. Each functor between layers is characterized by a universal arrow, which defines an isomorphism between every pair of categorical layers. These universal arrows define universal elements and representations through the Yoneda Lemma, and in turn lead to a new category of elements based on a construction introduced by Grothendieck. Causal inference between each pair of layers is defined as a lifting problem, a commutative diagram whose objects are categories, and whose morphisms are functors that are characterized as different types of fibrations. We defined causal effect in the UCLA framework using the notion of homotopy colimits associated with the nerve of a category. We illustrate the UCLA architecture using a diverse set of examples.

## Figures and Tables

**Figure 1 entropy-25-00574-f001:**
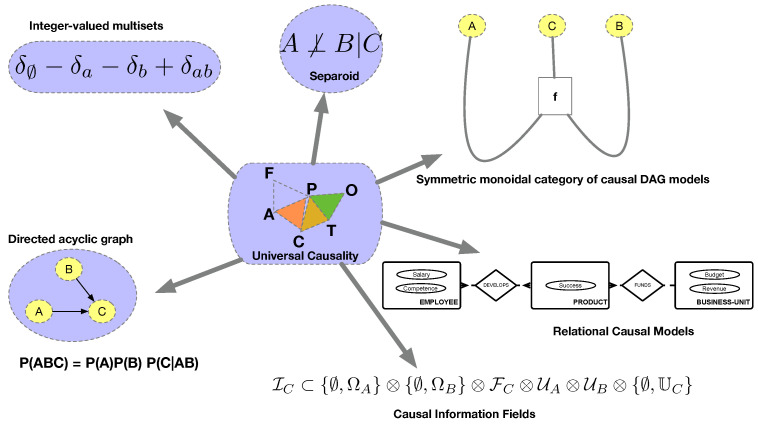
UC is a *representation-independent* framework that can be applied to many causal representations.

**Figure 2 entropy-25-00574-f002:**
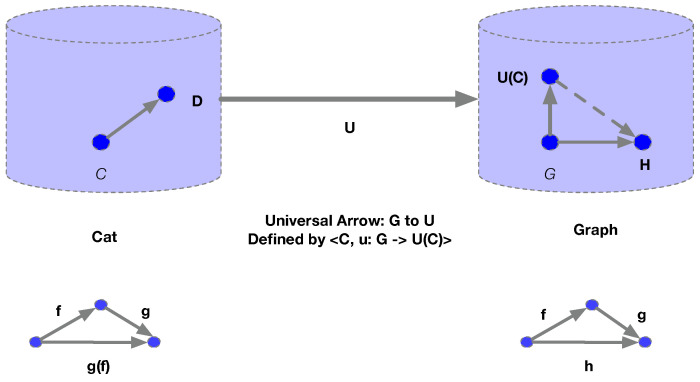
Universal arrows play a central role in the UCLA framework. In this example, the forgetful functor *U* between **Cat**, the category of all categories, and **Graph**, the category of all (directed) graphs, maps any category into its underlying graph, forgetting which arrows are primitive and which are compositional. The universal arrow from a graph *G* to the forgetful functor *U* is defined as a pair 〈C,u:G→U(C)〉, where *u* is a “universal” graph homomorphism. The universal arrow property asserts that every graph homomorphism ϕ:G→H uniquely factors through the universal graph homomorphism u:G→U(C), where U(C) is the graph induced by category *C* defining the universal arrow property. In other words, the associated *extension* problem of “completing” the triangle of graph homomorphisms in the category of **Graph** can be uniquely solved by “lifting” the associated category arrow h:C→D.

**Figure 3 entropy-25-00574-f003:**
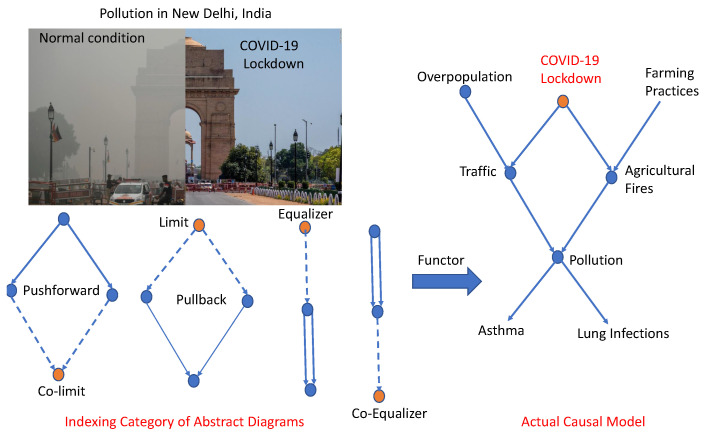
A causal model of climate change and COVID-19 lockdown. Universal causality defines causal models as functors mapping from an indexing category of abstract diagrams to the actual causal model.

**Figure 4 entropy-25-00574-f004:**
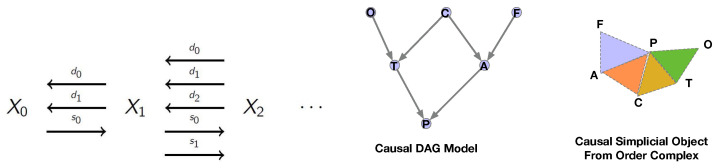
(**Left**) generic structure of a simplicial set. (**Right**) an oriented simplicial complex formed from the order complex of nonempty chains of the DAG model from Figure 3.

**Figure 5 entropy-25-00574-f005:**
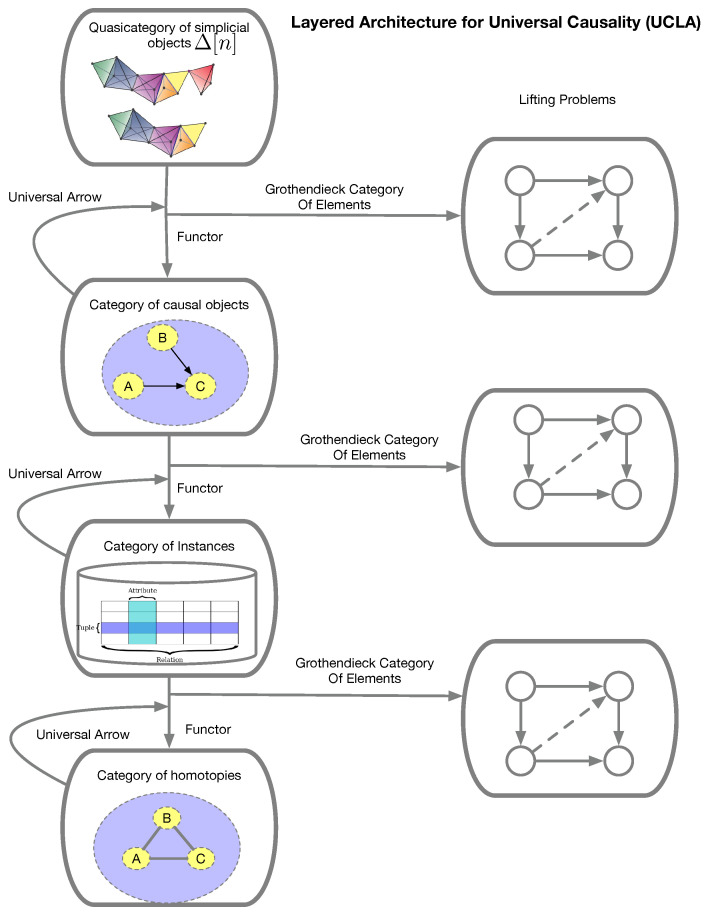
UCLA is a layered architecture that defines Universal Causality.

**Figure 6 entropy-25-00574-f006:**
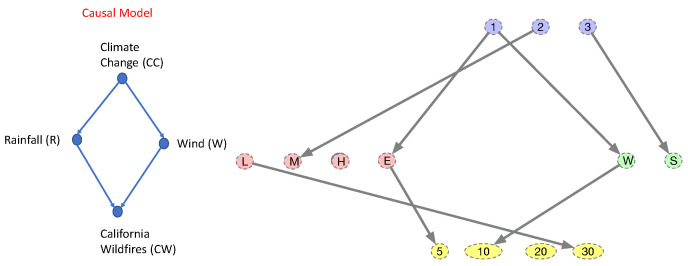
A toy causal DAG model of climate change to illustrate the category of elements construction. **Climate Change** is a discrete multinomial variable over three values 1,2, and 3. For each of its values, the arrow from **Climate Change** to **Rainfall** maps each specific value of **Climate Change** to a value of **Rainfall**, thereby indicating a causal effect of climate change on the amount of rainfall in California. **Rainfall** is also a multinomial discretized as low (marked “L”), medium (marked “M”), high (marked “H”), or extreme (marked “E”). **Wind** speeds are binned into two levels (marked “W” for weak, and “S” for strong). Finally, the percentage of California wildfires is binned between 5 and 30. Not all arrows in the category of elements are shown, for clarity.

**Figure 7 entropy-25-00574-f007:**
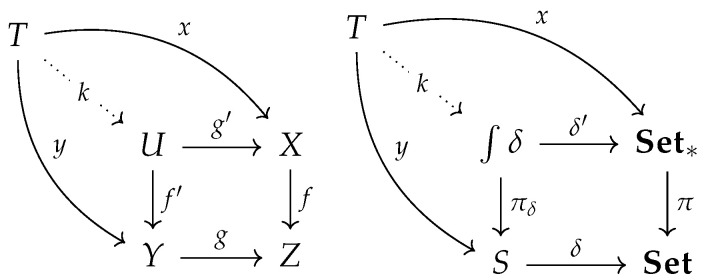
(**Left**) Universal Property of pullback mappings. (**Right**) The Grothendieck category of elements ∫δ of any set-valued functor δ:S→**Set** can be described as a pullback in the diagram of categories. Here, **Set**${}_{*}$ is the category of pointed sets (X,x∈X), and π is the “forgetful" functor that sends a pointed set (X,x∈X) into the underlying set *X*.

**Figure 8 entropy-25-00574-f008:**
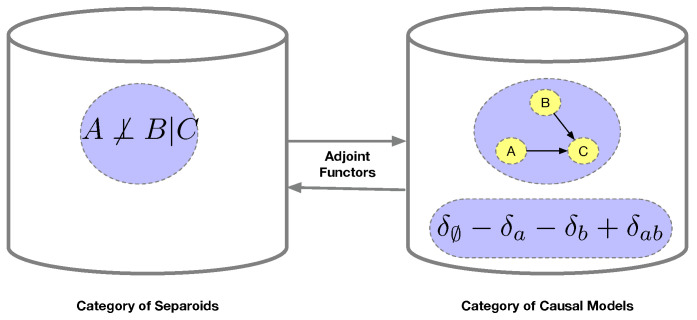
Adjoint functors between the category of separoids and the category of causal models. Here, a causal “collider" DAG over three random variables *A*, *B*, and *C*, and its associated integer-valued multiset, can both be viewed as “free" objects associated with a separoid conditional independence object, whereas the latter can be viewed in terms of a forgetful functor that throws away the causal DAG or integer-valued multiset structure.

**Figure 9 entropy-25-00574-f009:**
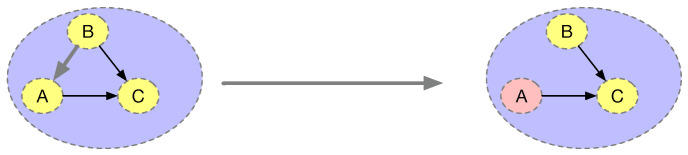
Causal interventions can be related to horns of a simplicial object.

**Figure 10 entropy-25-00574-f010:**
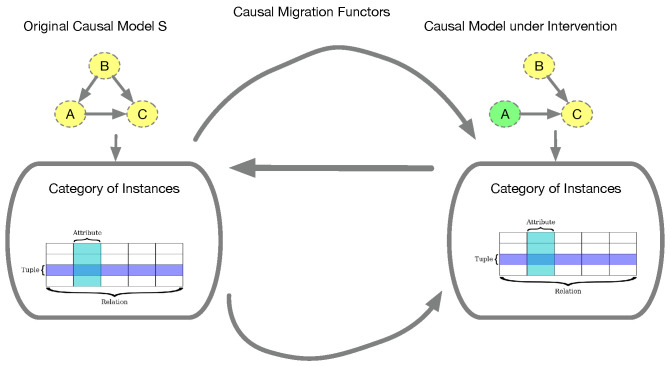
Kan extensions are useful in modeling the effects of a causal intervention, where in this example of a causal model over three objects A,B, and *C*, the object *A* is intervened upon, eliminating the morphism into it from object *B*.

**Table 1 entropy-25-00574-t001:** Category theory provides a unifying mathematical framework for relating the diverse formalisms used to study causal inference.

Representation	Objects	Morphisms	Citation
Rank-ordered statistics	Plants	Total ordering	Darwin [18]
Structural equation models	Variables	Algebraic equations	Wright [19]
Potential outcomes	Humans	Drug effects	Imbens and Rubin [9]
Directed Acyclic Graphs	Vertices	Paths	Pearl [5]
Distributive lattices	Subsets	Joins/Meets	Beerenwinkel et al. [20]
Relational causal models	Database schemas	Database relations	Maier et al. [21]
Information fields	Measurable Spaces	Measurable functions	Witsenhausen [6]
Resource Models	Monoidal resources	Profunctors	Fong and Spivak [1]
Universal Decision Models	UDM States	UDM morphisms	Mahadevan [22]
Counterfactual logic	Propositions	Proofs	Lewis [11]
Variational inequalities	Consumers/Producers	Trade	Nagurney [23]
Discourse sheaves	Users	Communication	Hansen and Ghrist [24]
String diagram surgery	Tensored objects	Tensored morphisms	Jacobs et al. [25]
Mean embeddings	RKHS embeddings	Mean maps	Muandet et al. [26]

**Table 2 entropy-25-00574-t002:** Each layer of UCLA represents a categorical abstraction of causal inference.

Layer	Objects	Morphisms	Description
Simplicial	[n]={0,1,…,n}	f=[m]→[n]	Category of interventions
Relational	Vertices *V*, Edges *E*	s,t:E→V	Causal Model Category
Tabular	Sets	Functions on sets f:S→T	Category of instances
Homotopy	Topological Spaces	Causal equivalence	Causal homotopy

## Data Availability

Not applicable.

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
