# Peer review of "Universal Causality"

_entropy, 2023, doi:10.3390/e25040574_

Round 1

Reviewer 1 Report (Previous Reviewer 2)

In the previous review I pointed to the lack of justification for using category theory for representing causal models. I feel this revised version fares even worse in this respect. The paper now contains more esoteric category-theoretic stuffs whose significance to the causal literature is unclear and almost never told explicitly. Another problem is that the paper is prohibitively difficult. I have basic knowledge of the theoretical machinery discussed in this paper (category theory, sheaf theory, Yoneda, Kan extension, etc.), but even with that it is painful to read it. I’m afraid very few people, if any, working in the causal literature fill find this paper helpful.

In sum, I have the following concerns which I fear make this paper not suitable for publication, at least in a theme issue on causality.

  • What are specific causal problems that can be solved by using this categorical framework? In other words, why do we need all this category theory?
  • The paper gives the impression of being just a survey of category theory & sheaf theory, or a list of definitions and expositions of known theorems. It is not clear at all why these results are relevant to the study of causation.
  • Mentions on the relevance to causal problems are very scarce and at most suggestive. Take, for instance, a passage around l. 325 where the author says “this process can be used to implement “surgery" of a causal model, such as a causal DAG,” but he does not tell us how and why. Likewise for all other parts.

It might well be possible that this paper is of interest to mathematicians working on category theory, but I’m afraid that it won’t appeal to causal theorists in general.

Author Response

Dear Reviewer, 

Thank you for your comments. In the revised submission, I have added a significant amount of explanation on why category theory provides a useful way to abstractly characterize causal inference. I hope these revisions will address your serious reservations. In particular: 

  1. I have included an example of work on relational causal models, and shown how the use of ground graphs in RCMs is related to the category of elements in UCLA. The discussion is brief as the goal of this paper is not a detailed study of relational causal models in category theory. 
  2. I have included an example showing how to define categories over conditional independence structures, in particular separoids, and show how we can abstractly model causal discovery using adjoint functors between the category of separoids and actual causal models. There's much more to be said here, but once again, the idea of using adjoint functors in relating conditional independence structures and actual causal models is the key contribution. 
  3. I have expanded the discussion of the background material, and included more explanation on higher-order category theory as providing a way to model causal interventions. I have added some references to more work in traditional causal inference, but also work across a wide spectrum of areas (Table 1). 

It is important to stress that UC is not intended to be a categorified causal DAG model, as recent work has already shown how to do this,  or even an extension of Pearl's d-separation in category theory. UC instead is primarily intended to be a conceptual framework showing how many of the specific structures used in causal inference can be understood by concepts such as universal arrows, which shows how to synchronize the morphisms over multiple categories (e.g., symmetric monoidal category models of Bayesian DAGs and stochastic matrices). UC also helps show connections to non-graphical representations, such as Witsenhausen's intrinsic model that led to causal information fields, and Studeny's integer-valued multisets that generalizes graph-based models. 

21st century mathematics is written in the language of category theory. A lot of work in graph-based causal inference uses concepts from 18th century mathematics (graphs, Bayes rule), and UC is an attempt to bring modern mathematics into the study of causal inference.  Categories are remarkably related to graphs, and yet bring in many new ideas. It is fairly easy to show that structural causal models, for instance, define categories, and that connection leads to many of the ideas in this paper. But, the role of the paper is not categorified structural causal models.  

Reviewer 2 Report (New Reviewer)

Entropy Review

Title: Universal Causality

A panoptic view of the author’s UCLA framework might be described like this: Each layer in the framework has (by some authors) been related to features of causal inference or modelling, while at the same time, since each layer is a well-defined category or class of categories, the relations (given by adjoint functors or universal arrows, which the author prefers) are equally well-defined, so we are in the happy position of being able to understand how these different categorial causal inferences fit together using category theory. Moreover, we can often lift certain constructions in one layer to another, allowing us to transfer or “mimic” (some of) our causal inferences between seemingly different domains of investigation. 

This seems correct to me, and so far as I can tell the author captures many of the interesting relationships between the layers they identify. Although I cannot be sure in all cases, I think the definitions they have provided are correct (there are a few little problems listed below). Moreover, I think this approach is useful and interesting. 

I have only one general criticism and many minor ones. The general criticism is roughly that, although the author claims to be offering an approach to understanding causation universally, the focus is rather more narrowly about causal graphs. They seem to take it to be enough – and for many, surely it will be enough – that they relate their approach to DAGs, the vogue models of causation and conditional independence a la Pearl (and I hasten to add, also analyzed by the philosopher Woodward). However, a reader interested in coming to conclusions or understanding about causation from a categorial perspective may be disappointed to see that the framework on offer seems, overall, to be an analytic tool for understanding different levels of categorial abstraction about DAGs. 

I say this is a general criticism, rather than a major one, because I do not think it requires a major revision. It is general because it permeates the article, but the author might handle it better simply by stating early on in the paper that they are considering causation from Pearl’s perspective, or so-called interventionism, or stating clearly which models of causation they will analyse. They should do this because there is no engagement with many other accounts or model of causation (e.g., no discussion of counterfactuals, or models given in the differential calculus). Indeed, there is nothing wrong with the paper on this account, besides that it seems to make more general claims (e.g. to being a “universal” account of causation), while only delivering an analysis of one corner of causal inference. Indeed, I understand that the prefix “universal” is intended by analogy with the universal mapping properties and universal constructions that are part of their approach, but it does suggest (along with the word universal in the layered architecture) that a more general treatment of causation is being offered.

The author may justifiably want to postpone a more broad investigation of different causal models in another paper. Indeed, in the future work section they say “One reason for pursuing this line of research is to expand the scope of causal models to beyond well-studied paradigms, such as Bayesian networks and causal DAGs.”. Although, it seems to me that the causal categories they do use within the paper are derived from DAGs, even if not themselves DAGs. Perhaps I am incorrect here. There are various parts of the paper wherein the author refers to causal categories, and perhaps they mean something like “Any category where the morphisms can plausibly be interpreted as causal relations”. If so, they should state this clearly. As it stands, the causal categories seem to be free categories generated by causal graphs.

Minor Comments:

1.     L.98. The “category of causal models” is mentioned, but not defined above. I am guessing that this is a category consisting of DAGs, or perhaps a categorification of DAGs, or perhaps it is the second layer, so it is what was called the “category of causal objects” above. Again on L.291, the reference is to a “causal category structure”, as layer 2. If all of these are the same, consider using the same term each time. This should be clarified. 

2.     The author should overall consider doing more to directly connect their points in abstract discussion to some features of causation. Although I am very sympathetic to the use of category theory in understanding and modelling causal concepts, I am not convinced from some of the author’s examples that they have identified the best way to do so, at least in some case. E.g. the example of “Causal Intervention and Horn Filling of Simplicial Complexes” the use of  horns of simplical complexes as models of forks in DAGs is positioned as an advantage of the author’s approach. However, when discussing how to understand the interventions on nodes in the DAG using this approach, the author claims that “A considerable elaboration of the theoretical machinery in category theory is required to describe the various solutions proposed” (l.371), and they do some of this elaboration in the following section. However, this leaves the reader wondering why they should use this approach, because it does not seem to require much advanced theory to classify or describe the various ways of filling out the original DAG from the example (indeed, the author lists them in a subsequent figure). So it is unclear what the benefit of the simplical approach is here. Perhaps, the author is caught in a bind, since they must use a simple example which does not properly illuminate the uses of simplicial complexes. However, I think I reader will want more here, and in many other cases, to convince themselves that there is causal knowledge gained from using these sorts of models. 

3.     L.146. Presheaves are seemingly identified with representable presheaves. But, not all presheaves are representable. Might want to note for the reader that Hom(-,X) type presheaves are a special class. Following paragraph goes into greater detail, but I still think it would be best to start with the usual definition of a presheaf as a functor C^{op} \to Set.

4.     L.148. If we are to consider causal diagrams in the categorial setting as diagrams (in the categorial sense of a functor from a small indexing category), then perhaps something should be said about the identity arrows, composition, etc. and their causal interpretation. I can imagine a sceptical reader objecting that the identity arrows seem to correspond to self-causes, or to a kind of trivial cause, so that this structure is not necessary and graphs, e.g. DAGs, are better. I can also imagine some scepticism about the composability of causes. If A causes B and B causes C, must there by any influence of A on C? Suppose I cause my partner to arrive at work on time, and my partner being at work on time causes them to read an email. It might nonetheless be the case that I have no influence on their reading the email (since they might have read it at home, or later in the day, etc.). Perhaps include some comment on the categorial meaning of causal terms, and why the diagrams approach is plausible.

5.     Similar point: in the move from a graph to a category via the free construction, what features are added to the graph might be significant and worth commenting on. E.g., even if the graph is not transitively closed, the underlying graph of the free category on that graph is necessarily transitively closed. This seems to wash away any intransitivity of causes. Might not be a problem for the author, but deserves note to the reader.

6.     I am confused by the definition of casually isomorphic change. L.197. How does the object X and X hat enter into the notion of change? Also, if we are to presume a notion of change for the definition, why is the definition not simply: “A causally isomorphic change is a change that is an isomorphism in the category whose morphisms are causal changes.”?

7.     L.307. A category S is mentioned, but not defined.

8.     L.331. There is a nice informal description of how to view the face operator in Example 7. However, it would be nice to have something like this description for the other examples as well.

9.     The author should elaborate a bit before the definition of the horn. L.350 It seems the notation \alpha([m]) has not been introduced. 

10.  Definition 3.8. Should there not be a continuity condition on the morphism? Is it lurking in the background somewhere?

11.  I think there is a mistake / typo in the definition of the objects of the category of elements, l.609.

12.  Example 14-15 seem to lack motivation. Example 14 claims that solving a lifting problem allows one (e.g.) to check whether a given set functor representing a causal graph will have a edge for each vertex, although for finite graph-like structures this would be easily checked without that method. Example 15 says the lifting approach allows one to check that two models have similar immoralities, although it would benefit the author if they could say something more about how/why this method is better. E.g., is it faster? Does it work for transfinite graphs?

13.  Def.4.5. Should mention that \phi is a natural isomorphism, not just a bijection.

14.  The causal reproducing property seems like a very important aspect of the author’s view, but it is not forecast or mentioned earlier in the paper. 

15.  Perhaps “Z(Sing2(X))” should not be repeated twice in the sequence in Example 16? Although I am no expert on singular homology.

16.  Even after reading about layer 4, I am unsure why precisely 4 layers are required. The author should consider mentioning why UCLA requires 4 layers, not 5+, or whether some cases 2 or 3 would be enough (or even better, perhaps). Overall, some further explanation of why the 4 layered architecture was chosen would help the paper.

I should say that I enjoyed reading the paper and found many interesting contributions in it, especially in the layered causal architecture. The author indeed makes their central point, that universal mapping properties relating these layers are potentially interesting and useful from a causal modellers perspective. 

Author Response

I thank the reviewer for the detailed comments on the paper. The revised submission contains many additional explanations that address your comments. Specifically, I will respond to some of your concerns below. 

"A panoptic view of the author’s UCLA framework might be described like this: Each layer in the framework has (by some authors) been related to features of causal inference or modelling, while at the same time, since each layer is a well-defined category or class of categories, the relations (given by adjoint functors or universal arrows, which the author prefers) are equally well-defined, so we are in the happy position of being able to understand how these different categorial causal inferences fit together using category theory. Moreover, we can often lift certain constructions in one layer to another, allowing us to transfer or “mimic” (some of) our causal inferences between seemingly different domains of investigation. "

Correct, except that the layers refer to particular problems within causal inference. So, Layer 1 uses ideas from higher-order category theory to define "intervention operators" that allow "surgery" of causal models, Layer 2 actually defines the causal model (such as a relational model), Layer 3 defines the dataset over which the model is defined, and Layer 4 defines equivalences among models using homotopy. 

"This seems correct to me, and so far as I can tell the author captures many of the interesting relationships between the layers they identify. Although I cannot be sure in all cases, I think the definitions they have provided are correct (there are a few little problems listed below). Moreover, I think this approach is useful and interesting. "

Thank you. 

"I have only one general criticism and many minor ones. The general criticism is roughly that, although the author claims to be offering an approach to understanding causation universally, the focus is rather more narrowly about causal graphs. They seem to take it to be enough – and for many, surely it will be enough – that they relate their approach to DAGs, the vogue models of causation and conditional independence a la Pearl (and I hasten to add, also analyzed by the philosopher Woodward). However, a reader interested in coming to conclusions or understanding about causation from a categorial perspective may be disappointed to see that the framework on offer seems, overall, to be an analytic tool for understanding different levels of categorial abstraction about DAGs. "

I have expanded the discussion to show UC applies to both graph-based and non-graphical models (see Figure 2 as an example). In particular, I have added a section on modeling conditional independence structures using separoids as a category, and shown how it can be related to causal models using adjoint functors. I have also included work on measure-theoretic causal models. 

"I say this is a general criticism, rather than a major one, because I do not think it requires a major revision. It is general because it permeates the article, but the author might handle it better simply by stating early on in the paper that they are considering causation from Pearl’s perspective, or so-called interventionism, or stating clearly which models of causation they will analyse. They should do this because there is no engagement with many other accounts or model of causation (e.g., no discussion of counterfactuals, or models given in the differential calculus). Indeed, there is nothing wrong with the paper on this account, besides that it seems to make more general claims (e.g. to being a “universal” account of causation), while only delivering an analysis of one corner of causal inference. Indeed, I understand that the prefix “universal” is intended by analogy with the universal mapping properties and universal constructions that are part of their approach, but it does suggest (along with the word universal in the layered architecture) that a more general treatment of causation is being offered."

This is a valid concern, and I did not want to include a detailed discussion of Lewis' work on counterfactual models in this paper, although I do believe UC applies to this work as well. It would require a substantial amount of discussion on how to model logic in category theory, and I felt it was a separate paper that I intend to write. There are other notions of causality as well (e.g., actual causality) that I do not address in this paper, which once again I hope to do in a future paper. 

"The author may justifiably want to postpone a more broad investigation of different causal models in another paper. Indeed, in the future work section they say “One reason for pursuing this line of research is to expand the scope of causal models to beyond well-studied paradigms, such as Bayesian networks and causal DAGs.”. Although, it seems to me that the causal categories they do use within the paper are derived from DAGs, even if not themselves DAGs. Perhaps I am incorrect here. There are various parts of the paper wherein the author refers to causal categories, and perhaps they mean something like “Any category where the morphisms can plausibly be interpreted as causal relations”. If so, they should state this clearly. As it stands, the causal categories seem to be free categories generated by causal graphs."

When I use the term "causal category", I mean it to imply any category structure that is intended to represent a causal model. This includes a vast array of work, for example structural equation models (SEMs) can be viewed as a category, as well as work on reproducing kernel Hilbert space models. Another example I do use in the paper is relational causal models, where the goal is to model entities in a relational database. 

Minor Comments:

"L.98. The “category of causal models” is mentioned, but not defined above. I am guessing that this is a category consisting of DAGs, or perhaps a categorification of DAGs, or perhaps it is the second layer, so it is what was called the “category of causal objects” above. Again on L.291, the reference is to a “causal category structure”, as layer 2. If all of these are the same, consider using the same term each time. This should be clarified."

Once again, a "causal category" is not a specific representation (like a DAG model), but intended to be a a class of representations that are defined as a category and intended to be used to define a causal model. For instance, one can define a causal category over measurable spaces and functions, which is exactly what causal information fields based on Witsenhausen's approach does. Similarly, one can define a category of SEM models, and so on. I expanded the discussion at the beginning (Figures 1 and 2) to emphasize this point. 

"The author should overall consider doing more to directly connect their points in abstract discussion to some features of causation. Although I am very sympathetic to the use of category theory in understanding and modelling causal concepts, I am not convinced from some of the author’s examples that they have identified the best way to do so, at least in some case. E.g. the example of “Causal Intervention and Horn Filling of Simplicial Complexes” the use of  horns of simplical complexes as models of forks in DAGs is positioned as an advantage of the author’s approach. However, when discussing how to understand the interventions on nodes in the DAG using this approach, the author claims that “A considerable elaboration of the theoretical machinery in category theory is required to describe the various solutions proposed” (l.371), and they do some of this elaboration in the following section. However, this leaves the reader wondering why they should use this approach, because it does not seem to require much advanced theory to classify or describe the various ways of filling out the original DAG from the example (indeed, the author lists them in a subsequent figure). So it is unclear what the benefit of the simplical approach is here. Perhaps, the author is caught in a bind, since they must use a simple example which does not properly illuminate the uses of simplicial complexes. However, I think I reader will want more here, and in many other cases, to convince themselves that there is causal knowledge gained from using these sorts of models. "

Simplicial complexes are interesting because they generalize directed graphs, which have been the main approach in Pearl's work. The notion of causality that emerges here is really higher-order causality, where instead of pairwise interactions among objects using an edge, you get interactions among groups of objects (an n-simplex represents the interactions n+1 objects, so a 1-simplex represents the traditional directed graph based causal model). I added more discussion of this point. 

"L.146. Presheaves are seemingly identified with representable presheaves. But, not all presheaves are representable. Might want to note for the reader that Hom(-,X) type presheaves are a special class. Following paragraph goes into greater detail, but I still think it would be best to start with the usual definition of a presheaf as a functor C^{op} \to Set."

Agreed, and I have added more explanation in this section. 

" If we are to consider causal diagrams in the categorial setting as diagrams (in the categorial sense of a functor from a small indexing category), then perhaps something should be said about the identity arrows, composition, etc. and their causal interpretation. I can imagine a sceptical reader objecting that the identity arrows seem to correspond to self-causes, or to a kind of trivial cause, so that this structure is not necessary and graphs, e.g. DAGs, are better. I can also imagine some scepticism about the composability of causes. If A causes B and B causes C, must there by anyinfluence of A on C? Suppose I cause my partner to arrive at work on time, and my partner being at work on time causes them to read an email. It might nonetheless be the case that I have no influence on their reading the email (since they might have read it at home, or later in the day, etc.). Perhaps include some comment on the categorial meaning of causal terms, and why the diagrams approach is plausible."

Compositionality is an interesting issue, and it is clear that often causality is compositional, even in the classic average treatment effect work of Rubin. Here, you model the effect of taking a drug on a patient in a clinical study. But, the drug needs to be ingested, it affects some particular part of the body, which in turn causes (or doesn't cause) a desired effect. All of this underlying biological machinery is ignored in the ATE literature, but it's really what drives the UC approach of explicitly modeling the compositional process the underlies causality. The Federal Reserve raises interest rates to try to  reduce inflation, and once again, the treatment here (raising interest rates) produces a desired effect (reduction of inflation) because of a whole host of intermediate cause-effect relations that are abstracted out (if interest rates go up, people are less inclined to buy houses or cars, which reduces their desirability and price etc.). So, much of what is traditionally modeled in the causality literature is compositional, except the standard approaches in economics (regression models) tend to obscure that a bit. 

As far as diagrams, the notion in UC is a bit more abstract than in traditional causal modeling. The idea in UC is very much in line with the way diagrams are viewed in category theory as functors from an indexing category to the actual category. 

"Similar point: in the move from a graph to a category via the free construction, what features are added to the graph might be significant and worth commenting on. E.g., even if the graph is not transitively closed, the underlying graph of the free category on that graph is necessarily transitively closed. This seems to wash away any intransitivity of causes. Might not be a problem for the author, but deserves note to the reader."

There is a notion of transitive DAGs, which leads to lattice conditional independence models. I refrained from a detailed discussion of this point as it would somewhat confuse the reader, but notice that graphs play multiple roles in the causality literature. The compositional nature of edges is what defines paths, which play a central role in Pearl's d-separation definition. There is a bit of slippery slope here, since some notions in d-separation are unrelated to causality (e.g., conditional independence is symmetric), but however play a key role in many causal discovery methods. I have added some discussion of this point. 

"I am confused by the definition of casually isomorphic change. L.197. How does the object X and X hat enter into the notion of change? Also, if we are to presume a notion of change for the definition, why is the definition not simply: “A causally isomorphic change is a change that is an isomorphism in the category whose morphisms are causal changes.”?"

What I was trying to do here is define a notion analogous to average treatment effect, where one makes an intervention on an object (e.g.., a patient takes a drug) and the goal is to see whether the treatment had an effect (e.g. X could be Federal Reserve raises interest rate and Y would be inflation is reduced by some percent). In Pearl's approach, ATE is replaced by the do-calculus formalism. In one case, you measure causality as the mean shift in a distribution, and the other measures causality in terms of whether the observational and interventional distributions ar the same. One can even use other definitions, e.g. Janzing et al. in Annals of Statistics define causal effects in terms of the KL-divergence between the two situations (pre and post intervention). In my definition, I was trying to define change in terms of category theoretic notions like isomorphism between two objects. But I like your revised definition, and included it as a footnote as an alternative and perhaps simpler definition.  

"L.307. A category S is mentioned, but not defined." 

This refers to a causal category. 

"L.331. There is a nice informal description of how to view the face operator in Example 7. However, it would be nice to have something like this description for the other examples as well."

Added similar descriptions to some other examples as well. 

"The author should elaborate a bit before the definition of the horn. L.350 It seems the notation \alpha([m]) has not been introduced. "

Note \alpha is a morphism in the set Hom([m], [n]), so \alpha([m]) refers to its co-domain value corresponding to domain element [m]. 

"Definition 3.8. Should there not be a continuity condition on the morphism? Is it lurking in the background somewhere?"

In defining the homotopy between continuous functions $f$ and $g$ in topology, one imposes a continuity condition of the form H: X \times [0,1] \rightarrow Y$, which restricted to 0 gives us a function $f$ and $g$ when restricted to 1. Here, we replace the condition in topology with H: X \times \Delta_1 \rightarrow Y$, where $f$ and $g$ are maps between simplicial sets $X$ and $Y$, which are produced by restrictions $i_0, i_1: \Delta_0 \rightarrow \Delta_1$, where $i_j(0) = j$ induces a map $\Delta_0 \rightarrow \Delta_1$ (Page 225, Section 10.5 of the book From Categories to Homotopy Theory by Richter, Cambridge). 

"I think there is a mistake / typo in the definition of the objects of the category of elements, l.609." 

Typo fixed, thank you! 

"Example 14-15 seem to lack motivation. Example 14 claims that solving a lifting problem allows one (e.g.) to check whether a given set functor representing a causal graph will have a edge for each vertex, although for finite graph-like structures this would be easily checked without that method. Example 15 says the lifting approach allows one to check that two models have similar immoralities, although it would benefit the author if they could say something more about how/why this method is better. E.g., is it faster? Does it work for transfinite graphs?"

Your criticism is valid, but the purpose of these examples was to illustrate the use of lifting problems for posing graph problems that are relevant to causal inference, and not to suggest novel algorithms (although that is certainly an interesting question for future work!). The paper is long enough already without delving into the issues of designing efficient inference algorithms. 

"Def.4.5. Should mention that \phi is a natural isomorphism, not just a bijection."

Corrected, thank you. 

"The causal reproducing property seems like a very important aspect of the author’s view, but it is not forecast or mentioned earlier in the paper. "

I have moved the discussion of CRP to an earlier section of the paper, thank you. 

"Perhaps “Z(Sing2(X))” should not be repeated twice in the sequence in Example 16? Although I am no expert on singular homology."

That was a typo, corrected! 

"Even after reading about layer 4, I am unsure why precisely 4 layers are required. The author should consider mentioning why UCLA requires 4 layers, not 5+, or whether some cases 2 or 3 would be enough (or even better, perhaps). Overall, some further explanation of why the 4 layered architecture was chosen would help the paper."

I don't think 4 layers are necessary always, as functors do compose, and often one can get by with 3 layers or fewer, at the expense of being able to do fewer things. For example, in the work on encoding causal DAGs as symmetric monoidal categories using string diagram surgery, only two layers are used (one layer defines the SMC model, and the other layer encodes the stochastic matrices defining the model). 

The important point I wanted to emphasize was that causal inference is best thought of using a layered categorical structure, and most importantly, the causal interventions themselves should be thought of as defining a category (as interventions naturally compose, and most work in causal discovery in fact uses a planned sequence of interventions to isolate a particular causal model). 

Thank you for your detailed comments, they were immensely helpful. I hope the revised paper addresses most of your major concerns. 

Reviewer 3 Report (New Reviewer)

The paper is technically sound.  Words like 'Yoneda' and 'presheaf' are used correctly.  

But I am unable to determine if this venue is appropriate for the paper.  The paper strikes me as mostly observational in nature, which could be appropriate for some venues (e.g., many math journals), but not others (e.g. many computer science journals).  It could be that the paper's observations are of great importance, but I can't judge that either: the reason is, despite being a specialist in the area, I don't know how to implement the mathematics in the paper on a computer, a necessary prerequisite for me to judge utility (I think utility derives from getting computers to do things they couldn't previously do, or do faster); for me to implement on a computer, the paper's math must be phrased in terms of presentations and algorithms ('Algebraic Data Integration' in the Journal of Functional Programming does this well); so much of category theory is infinitary and how to perform it on a computer is not obvious.   For example, I got really excited about the possible existence of "probabilistic functorial data migration" until it became apparent I had no clue how to implement such a thing based on the paper alone.

In addition, the paper would benefit from more examples (there are few, and they don't span the entire paper or integrate well with each other) as well as an argument for why the paper's observations are important - what new capability does the paper allow, if any?  

Finally, the reader pre-requisites need to be clarified: any reader capable of following the category theory would probably not need to be introduced to concepts such as 'pushout' as though they are new, and the paper should simply be honest about requiring graduate level category theory to understand (which also makes me wonder if a math journal would be a better venue, but I'm unfamiliar with Entropy).

Author Response

Dear Reviewer, 

Thank you for your comments.

"But I am unable to determine if this venue is appropriate for the paper. "

" I don't know how to implement the mathematics in the paper on a computer, a necessary prerequisite for me to judge utility" 

Your concerns are valid in that the role of this paper is not to propose a specific computer implementation of causal inference using a specific representation.  There are plenty of such papers that have already been published in the past few decades. What the paper tries to do is illuminate "universal properties" that underlie causal inference, such as the causal reproducing property, the role of universal arrows in causal inference, and the usefulness of defining causal interventions themselves as a category using simplicial sets and objects. 

It is certainly my goal to translate the ideas in this paper into practical algorithms for particular datasets in future papers. For example, the connection to relational causal models, which I have added to this paper, is such an extension that would be best explored in a subsequent paper, since it would require building in additional specific assumptions that are specific to modeling relational causal models. 

I was invited by the editor to submit to this special issue of Entropy on causality, so I have to hope and assume that this particular style of research in causal inference is suitable to the journal! 

"In addition, the paper would benefit from more examples (there are few, and they don't span the entire paper or integrate well with each other) as well as an argument for why the paper's observations are important - what new capability does the paper allow, if any? "

I have added some more examples in the revised paper, such as relational causal models. The new capabilities that emerge come from thinking of causal inference in terms of simplicial sets and objects, which are generalizations of directed graphs as well as ordinary categories. So, for example, in graph-based models, we think of pairwise interactions among objects. In simplicial sets, we can model interactions among an arbitrary number of objects in an n-simplex. Similarly, we can do causal interventions on groups of objects simultaneously, which as I note in the paper, may be useful in causal inference under interference. 

"Finally, the reader pre-requisites need to be clarified: any reader capable of following the category theory would probably not need to be introduced to concepts such as 'pushout' as though they are new, and the paper should simply be honest about requiring graduate level category theory to understand (which also makes me wonder if a math journal would be a better venue, but I'm unfamiliar with Entropy)."

I tried to include some background material on category theory, with the emphasis being on how these were useful in causal inference. I have added some more explanation in this revised submission. Certainly, the paper would be a lot shorter if I took out all background material, but then the paper would have potentially a much narrower audience. 

Thank you again for your comments. 

Round 2

Reviewer 2 Report (New Reviewer)

I believe the author has taken every opportunity to improve the manuscript and has clearly taken into account each of the major and minor concerns I raised in my original review. I still have slight reservations about the informal justifications associated with using the approach in specific cases of causal inference, and some concerns about the assumptions of transitivity of causal relations. However, the author is correct that causation is transitive, or taken to be so when modelled, in many interesting cases, to which their sort of approach will apply. I think the approach suggested is powerful and interesting, and hope to see some interesting applications of it in future works. 

Reviewer 3 Report (New Reviewer)

The paper is fine - my two objections have been addressed.  The article is invited, which addresses my concern about being an appropriate venue.  The author's reply states that computability isn't a concern, so my other objection has also been addressed (of course, a comment to the effect that computability is being deliberately ignored for expediency would be welcome).

This manuscript is a resubmission of an earlier submission. The following is a list of the peer review reports and author responses from that submission.

Round 1

Reviewer 1 Report

Thank you for an interesting paper. In an effort to be comprehensive, I feel you have strayed too far from your paper’s objectives. By including so many famous names, for example, Line 172 and following mentions Descartes, Hume, Darwin, Galton, Fisher, Sewall Wright, Pearl, you detract from your own technical work.  Line 96 says much of what you need to say on these broader aspects.

Author Response

Dear Reviewer, 

Thank you for your detailed comments. As I note in my cover letter for the revised paper, the paper has been completely rewritten, and is in many ways a new paper, albeit with the same motivation and themes. To address your specific points for the original paper, here are some brief comments: 

  1. "Mathematical abstraction of Plato" and philosophical comments: These have been completely removed from the paper, and I agree do not belong here.
  2. Yoneda Lemma: The analogy to physics has been deleted. 
  3. Title of Figure 1: All figures have been changed, and there is now a new Figure 1 showing the proposed layered architecture for causal reasoning. 
  4. Lewis counterfactuals: this has been deleted as a full discussion of Lewis' work would distract from my paper. 
  5. d-separation discussion has been deleted. 

Reviewer 2 Report

This manuscript aims to formulate the Pearl-style causal models with the language of category theory. In particular, relying on results from sheaf theory, the author claims to provide two universal properties of causal theory, respectively dubbed Universal Causal Theorem (UCT) and Causal Reproducing Property (CRP).

Since the connection between the causal theory and category theory is a promising area with much theoretical potential, the present manuscript raises high expectations. But after reading the paper I have to conclude that it fails to meet the expectations it promises. I think the paper as it stands now has the following issues.

1. The paper lacks justification for its category-theoretic formulation of causality. Below are some samples where I found further justification or explanation is needed.

  • The author seems to presume that edges in a causal model correspond to morphisms in a category (l. 260-264). But causal edges are not necessarily composable. A→B and B→C do not imply the existence of A→C in general.
  • It is said that “confounding implies that the presheaf functor Hom_C(-,X) cannot be fully determined” (282-3), but what does it mean? If it just means that a confounding factor is not in the set of known causes of X, it is rather a truism, and also, what it explains is just that it is unobserved and not that it is a confounder (non confounding unobserved cause of X is not in the homset either).
  • The author identifies causal intervention with an arrow from a unit object 1 (338). But why is this an intervention? There are two issues. First, in category Sets, such an arrow would pick up an element. But we are not told almost anything about the nature of objects, so it is not clear what is the nature of 1→X (in 551, it is said the category DAG has its objects directed acyclic graphs. But this should be an error. Maybe variables? But even still, we are not told what they are). Second, it is unclear why 1→X serves as an intervention. Even with an analogy with Sets, an intervention should be much more than identifying a single element or value.
  • Around 407, the factorization property of Bayesian networks is said to have “the strong resemblance” to the property of sheaf. But I don’t see the resemblance. And is it just a resemblance, or something that has a theoretical significance? (for instance, can we say that the unfactorizable distribution fails to be a sheaf, is so why?) This clearly needs more explanation.
  • The paper mostly discusses the graphical properties (DAG) of causal models, but not much associated probability distribution. The connection to the probabilities or structural equations is briefly mentioned around definition 3.1, when the author discusses the evaluation functor. But clearly more must be said about the nature of this functor, and the category to which it sends C.

2. The main theorems of this paper, the UCT and CRP, are just restatements of the well-known classical results in sheaf theory. What makes these results “causal” is just the assumption that the basis homsets represent causal influence. This assumption, however, is not well established as discussed above. Also, even if we grant that the homset Hom(-,X) does represent causes of X, the significance of the theorems is not clear. For instance, the UCT claims that any such homset is representable as a colimit of a certain diagram in a presheaf category. But what does it mean for the study of causality? What question does it solve? Similar concerns apply to other “results” of the paper, including the CRP and Kan extensions. To convince a reader of the usefulness of category theory the author needs to show which questions these results are meant to address. Without that, all these categorical talks sound like abstract nonsense (which, of course, is an oft-leveled criticism against category theory).

Finally, just FYI, Jacobs et al. (2019) have recently given an elegant application of category theory and especially monoidal categories to causal modeling.

Jacobs, B., Kissinger, A. & Zanasi, F. Causal Inference by String Diagram Surgery. in Foundations of Software Science and Computation Structures 313–329 (Springer International Publishing, 2019).

Author Response

Dear Reviewer ,

Thank you for your detailed comments, and for the reference to Jacobs et al. work on symmetric monoidal category representations of causal Bayesian DAG models. As you will note from the revised paper, the original paper has been completely rewritten, and in many ways it is an entirely new paper, albeit with the same motivation and themes. I detail the major changes in the letter to the editor and to all the reviewers that accompanies my submission. I will focus my response to your comments on the original paper, noting that in many cases, the paper's revisions adds significant new detail. 

  1. "Causal edges are not necessarily composable:": this raises an interesting question. In the causal graphical models literature, graph edges are composed into paths, which is an essential aspect of the definition of d-separation and conditional independence. There is an intrinsic notion of composition in causality (e.g., if the massive amount of Covid-19 relief package signed by Congress increased the inflation rate of the US economy, and the high inflation rate of the US economy caused prices to increase, we would tend to view the Covid-19 relief package as at least an indirect cause of higher prices). Without getting into a long discussion, there is a natural connection between graphs and categories, one that I discuss in detail in the revised paper through the notion of universal arrows. Perhaps this may clarify this important question, one that I have thought about a lot as well!
  2. Confounding is alas an important issue that I regretfully had to omit discussion of in the revised paper, even though it is an important topic in causality. Fundamentally, confounding is treated in the literature through statistical analyses (e.g., Rubin treats it as a missing value problem), and my goal was to stay within pure category theory as much as possible in defining universal causality. To answer your excellent question here, the way many researchers have approached confounding in graphical models approaches to causality is to use undirected edge (dotted lines in Pearl's book) or hyper edges. The assumption is that even though the confounders are not observable, their effects on the observed variables are. This raises many issues, addressing which would detract from and lengthen an already long revised paper! It is another question that I have spent a lot of time thinking about! There are various ways to model confounding in a category-theoretic way, but I could not focus on any one in sufficient detail, so I had to reluctantly postpone the discussion to a future paper. I note that Jacobs et al. do address a bit on confounding, and their approach is certainly consistent with my proposed UCLA hierarchy. 
  3. Interventions are now modeled using the category of simplicial objects, so the revised paper proposes a different approach to interventions. There are other ways to model it, e.g. Jacobs et al. proposes using endofunctors, whereas Spivak proposes using migration functors in his work on database functorial migration. There are many ways to deal with interventions, and I hope the revised paper will address your question better. 
  4. Regarding sheaves, there is a whole paper waiting to be written on how the theory of sheaves plays a key role in causal inference, and once again, I could not do justice to this topic in this paper! Very briefly, key in the definition of sheaves is the notion of ensuring that the restriction of a function (say defined on the open sets of a topology) to a subset of an open set will be consistent. That is, if an element is in the intersection of two open sets, and each has a function defined over it, the intersection of these two function definitions must be consistent. In casual inference, there is an analogous situation when a model is defined combinatorially as an algebraic structure made up of parts. In graphical models, the sheaf condition is satisfied because conditioning on the model produces a submode that is consistent. There is a more general way to specify this with sheaves, but I realized that getting into the details of sheaves would distract from the revised paper. I note briefly at the end that Grothendieck topology (which specifies a topology on any category) is a useful way to specify structure on a causal model in a purely categorical way, and I hope to write a further paper on causal inference over sheaves on sites, but that would not be possible in this paper. 
  5. I focused largely on DAG models, but not on probability distributions, and that holds true for the revised paper as well. In part, what I wanted to do is define causality purely category theoretically, and introducing probability seemed like a special case. In the literature on causal inference, researchers use probabilistic models, Hilbert space models, logic, regression models and many other representations. My original goal was to define causality purely in a representation-independent manner. I don't know if I succeeded in that in this paper, since  I define causal effects towards the end using the associated homotopic structure on a causal model from its topological realization. For many people, it might be preferable to define causality using paobabitlies (but this leads to awkward problems as Bayes rule inverts arrows and then many graphical models become non-identifiable from purely observational data). It's a problem I continue to think about a lot, but in this paper, I don't include a substantial discussion of probabilistic representations. Of course, Jacobs et al. use stochastic matrices as a category to define causal models (which is the second layer and third layer in the UCLA hierarchy).  Their approach is certainly fine, but Witsenhausen's approach uses measure theory (which Heymann et al. used in their work on causal information fields) and I think Witsenhausen's approach is more general than the one proposed by Jacobs et al. It's still an open problem on what is the best way to model causality in terms of imposing a structure on the data that is used to build the model. 
  6. The discussion of Yoneda Lemma and the density theory of sheaves has now been altered, because I have chosen to present them in the context of the universal arrow property (which is how Maclane presents the Yoneda Lemma, and I like this because it clarifies for me also the connection between category structures and directed graph structures). Ultimately, it seems to. me that causal influences can only be transmitted through morphisms in a category (e.g., in Jacob et al. work, the morphisms of the SMC CDU category convey causal information), and although Jacobs et al. don't mention the Yoneda Lemma or pre sheaves, it is fundamental theorem of category theory that any object is defined up to isomorphism through its presheaf. So, I continue to believe that presheaves ultimately are the carrier of causal information, in that if an intervention is done, the presheaf must change (in fact, I note that Grothendieck's topology is defined with respect to the presheaf, and that's another way to formalize this notion, although I didn't have the space to do it in this paper). 

Once again, I want to thank you for your many insightful comments, and for the reference to Jacob et al. work, which I have benefited from reading. Our goals are different, as I am trying to define causality in a more general way than they are, and my goal is not to "categorify" Bayesian network DAG models (which Fong did as well in his MS thesis). I do not know if I have succeeded in convincing you in the revised paper, but I hope the substantially revised paper will have addressed some of your major concerns. 

There is much detail that had to be regretfully omitted, and will be in future papers. I note that I have written two other papers in the interim, one on categoroids ( modeling conditional independence in a category) and another paper on using lifting problems in higher order category theory in causal inference. Some of the ideas in those papers crept in here, but there is a more substantial paper on causal discovery that I plan to write based on these ideas, which has to be another paper entirely. 

Reviewer 3 Report

The author proposes a mathematical framework for causality and causality inference based on category theory. The paper focuses on DAG-based models of causality, in the tradition of Pearl. The main idea is to consider causal diagrams as diagrams in the sense of category theory: functors out of small categories (the indexing categories) into a given category of interest. To an applied category theorist like myself, this idea should seem natural enough. Thus, the premise of the paper is at least plausible.

Unfortunately, the idea is not executed in a satisfactory manner within this paper, which suffers from at least two major problems.

First, the paper makes broad and sweeping claims about its results that are not substantiated. This tendency is apparent from the opening sentences of the manuscript, where it is claimed that the proposed approach unifies the theories of causality due to Plato and Pearl. However, no mention of Plato is made in the rest of the paper. Later in the Introduction, it is claimed that "UC [universal causality] can be viewed as a “Rosetta Stone" [20] to translate across diverse representations used in causal inference." Although allusions are made to various causal representations, it is never clarified how UC encompasses or unifies them. Such overly vague and unsupported claims appear throughout the paper.

Second, and most importantly, the attention to mathematical detail and correctness is simply not adequate for a contribution to the mathematical theory of causality. Many definitions and statement involve mathematical errors or are too imprecise to be mathematical statements at all. For example, the definition of the category DAG on page 10 is ill-specified---the notion of morphism does not make sense---and the reference to enriched category theory there is both unnecessary and garbled. Crucially, what should be the main definition of the paper, Definition 3.1 of a Universal Causal Model, contains errors and is not a well-specified mathematical definition. Moreover, a number of claimed examples of the definition are included in the definition itself, so it is not even clear what parts are definitions or examples.

The material in this paper would be better served by a more focused and careful study. It cannot accomplish its goals in its present form.

Author Response

Dear Reviewer,

I thank you for your concise and succinct review, which I took to heart. The revised paper has been substantially rewritten and I hope addresses many of your entirely valid concerns with the original paper. I introduce a much more detailed hierarchical model of causal inference using several layers of categorical abstraction. In the letter to the editor and the reviewers, I note the major changes to the paper, and in many ways, it is an entirely new paper, one that I hope addresses at least some of your major concerns. 

  1. The connection between the notion of a morphism in category theory and an edge in a DAG is an interesting problem, one that I have spent a lot of time thinking about. As shown in Maclane's book and many others, directed graphs are essentially categories "without a composition rule", and in the revised paper, I use the notion of universal arrow that Maclane uses in his book as a motif to elucidate the connection between edges in a graph and morphisms in a category.
  2. Category theory primarily defines notions using universal properties, and in the revised paper, pullbacks and lifting diagrams play a key role. These hopefully will clarify what I was trying to get at in the original paper. I do believe that the use of lifting problems at various levels of the UCLA hierarchy clarify how these universal constructions in category theory are useful in modeling causal inference, and I hope the additional details will address your valid concerns. 
  3. Categories can be defined in countless ways, and morphisms in a category can mean substantially different things depending on what the underlying category is. For example, the category of all directed graphs has as its morphism the graph homomorphism itself. In the work of Fong and Jacobs et al. on symmetric monoidal categories encoding causal Bayesian networks, the morphisms of the CDU category relate to the edges of the DAG quite directly -- as the DAG structure changes, the morphisms change. In my ULCA hierarchy, there are several category structures defined, and the morphisms accordingly play many roles. At the simplicial layer, morphisms are defined for doing ``surgery" by extracting parts out of a causal model, and are order preserving maps on the ordinal numbers. Following Spivak's use of relational databases as categories, one can treat morphisms as he does as column headers in tables. In the work on integer-valued multisets by Studeny, which I discuss in the revised paper, morphisms mean something else again. 
  4. If you look at the literature on causal inference, I think it's fair to say there is no agreement on what is causality, and many approaches have been pursued over the past 60 years or more. Economists define causality in terms of regression models, Pearl defines it in terms of a directed acyclic graph, Rubin and other statisticians define it in terms of average treatment effect estimation. I think it would be too challenging to say there must be one definition of causality in category theory that would satisfy everyone. What I tried to do in the revised paper was elucidate a framework that may be useful in defining causal inference in a number of different ways, depending on the particular ways in which the architecture is realized. Thus, if one uses Bayesian DAG models, one can certainly "plug in" Jacobs et al. or Fong's encoding of DAG models as symmetric monodical categories in layer 2 of the UCLA hierarchy. That would not cover other approaches, such as integer-valued multisets, or even the work on Hilbert-space embedding of average treatment effect models (e.g.. the work on counterfactual mean embedding). In this paper, I did not include the definition of a universal causal model that I attempted to do in the first paper, for these reasons. I do believe that there is value to defining an architecture at some sufficient level of generality, which can be used to implement different finer grained notions of causality (much as categories themselves are defined in a way general enough to permit countless instantiations). 

Thank you again for your review, and I do hope the revisions address some of your major concerns. Working on this paper has been an exciting intellectual adventure for me, and as someone not an expert in category theory, it has been an educational one as well. I am sure the paper can benefit from further revisions, and I hope that in the fullness of time, I will be able to nail down an approach that I find satisfactory to me and that addresses your primary concerns as well.